# Flavodiiron proteins 1–to-4 function in versatile combinations in O$_2$ photoreduction in cyanobacteria

Anita Santana-Sanchez, Daniel Solymosi, Henna Mustila[†], Luca Bersanini[‡], Eva-Mari Aro*, Yagut Allahverdiyeva*

Molecular Plant Biology, Department of Biochemistry, University of Turku, Turku, Finland

**\*For correspondence:**
evaaro@utu.fi (E-MA);
allahve@utu.fi (YA)

**Present address:** [†]Department of Chemistry–Ångström Laboratory, Uppsala University, Uppsala, Sweden; [‡]Biophysics of Photosynthesis, Department of Physics and Astronomy, Vrije Universiteit Amsterdam, Amsterdam, The Netherlands

**Competing interests:** The authors declare that no competing interests exist.

**Abstract** Flavodiiron proteins (FDPs) constitute a group of modular enzymes widespread in Bacteria, Archaea and Eukarya. *Synechocystis* sp. PCC 6803 has four FDPs (Flv1-4), which are essential for the photoprotection of photosynthesis. A direct comparison of light-induced O$_2$ reduction (Mehler-like reaction) under high (3% CO$_2$, HC) and low (air level CO$_2$, LC) inorganic carbon conditions demonstrated that the Flv1/Flv3 heterodimer is solely responsible for an efficient steady-state O$_2$ photoreduction under HC, with *flv2* and *flv4* expression strongly down-regulated. Conversely, under LC conditions, Flv1/Flv3 acts only as a transient electron sink, due to the competing withdrawal of electrons by the highly induced NDH-1 complex. Further, in vivo evidence is provided indicating that Flv2/Flv4 contributes to the Mehler-like reaction when naturally expressed under LC conditions, or, when artificially overexpressed under HC. The O$_2$ photoreduction driven by Flv2/Flv4 occurs down-stream of PSI in a coordinated manner with Flv1/Flv3 and supports slow and steady-state O$_2$ photoreduction.
DOI: https://doi.org/10.7554/eLife.45766.001

## Introduction

A-type flavodiiron proteins (Flvs or FDPs) were originally identified in strict and facultative anaerobes among Bacteria, Archaea and Protozoa and were considered to function in O$_2$ and/or NO detoxification (*Wasserfallen et al., 1998*; *Gonçalves et al., 2011*; *Folgosa et al., 2018*). All FDPs share two conserved structural domains: the N-terminal metallo-$\beta$-lactamase-like domain, harboring a non-heme diiron center, where O$_2$ and/or NO reduction takes place; and the C-terminal flavodoxin-like domain, containing a flavin mononucleotide (FMN) moiety. The structures of FDPs in anaerobic prokaryotes and eukaryotic protozoa have been resolved as homooligomers (dimer or tetramer comprised of two dimers) arranged in a 'head-to-tail' configuration, so that the diiron center of one monomer and the FMN of the other monomer are in close proximity to each other, which ensures rapid electron transfer between the two cofactors.

C-type FDPs, specific to oxygenic photosynthetic organisms, hold an additional flavin-reductase-like domain, coupled with extra cofactors (*Romão et al., 2016*; *Folgosa et al., 2018*). *Synechocystis* sp. PCC 6803 (hereafter, *Synechocystis*) possesses four genes encoding FDPs: *sll1521* (Flv1), *sll0219* (Flv2), *sll0550* (Flv3) and *sll0217* (Flv4). Recently resolved crystal structure of truncated Flv1 from *Synechocystis* revealed a monomeric form with a 'bent' configuration, however the organization of the additional flavin-reductase-like domain and the oligomeric structure remain unclear (*Borges et al., 2019*). Photosynthetic FDPs first gained attention in 2002, when recombinant *Synechocystis* Flv3 protein was shown to function in O$_2$ reduction to water without producing ROS (*Vicente et al., 2002*). Later, it was demonstrated that *Synechocystis* Flv1 and Flv3 proteins function in vivo in the photoreduction of O$_2$ downstream of Photosystem (PS) I (*Helman et al., 2003*). Since then,

extensive research has been performed to reveal the crucial function of Flv1 and Flv3 (and their homologs, FLVA and FLVB in other photosynthetic organisms) as a powerful sink of excess photosynthetic electrons. This safeguards PSI and secures the survival of oxygenic photosynthetic organisms under fluctuating light intensities (*Allahverdiyeva et al., 2013*; *Gerotto et al., 2016*; *Chaux et al., 2017*; *Jokel et al., 2018*) or under short repetitive saturating pulses (*Shimakawa et al., 2017*). The Flv1- and Flv3-mediated light-induced alternative electron transport to $O_2$ was named as the Mehler-like reaction, being a widespread pathway, operating in nearly all photosynthetic organisms from cyanobacteria up to gymnosperms, but lost in angiosperms (*Allahverdiyeva et al., 2015*; *Ilík et al., 2017*).

The Flv2 and Flv4 proteins are encoded by an operon, together with a small membrane protein, Sll0218. The *flv4-sll0218-flv2* (hereafter *flv4-2*) operon is strongly induced in low inorganic carbon, $C_i$, (atmospheric 0.04% $CO_2$ in air, LC) and high light conditions (*Zhang et al., 2009*). The operon structure is highly conserved in the genome of many $\beta$-cyanobacteria (*Zhang et al., 2012*; *Bersanini et al., 2014*). The *flv4-2* operon-encoded proteins have been reported to function in photoprotection of PSII by acting as an electron sink, presumably transporting electrons from PSII or the plastoquinone (PQ) pool to an unknown acceptor (*Zhang et al., 2009*; *Zhang et al., 2012*; *Bersanini et al., 2014*; *Chukhutsina et al., 2015*). Since *flv2*, *sll0218* and *flv4* are co-transcribed, the contribution of each single protein of the operon to PSII photoprotection has been difficult to dissect. Recent data examining distinct and specific roles of the Flv2/Flv4 heterodimer and the Sll0218 protein (using a set of different mutants deficient only in Sll0218 or in Flv2 and Flv4) demonstrated that the majority of observed PSII phenotypes were actually due to the absence of Sll0218, thus leading to the conclusion that Sll0218 contributes to PSII repair and stability (*Bersanini et al., 2017*). However, the exact donor and acceptor of the Flv2 and Flv4 proteins have not yet been identified in vivo and possible cross-talk between all four FDPs has yet to be revealed, thus limiting our understanding of the function of FDPs on a cellular level.

In this work, to shed light on the in vivo function of Flv2 and Flv4 and to clearly separate the function of the Flv1/Flv3 heterooligomer from that of Flv2/Flv4, we employed a specific set of FDP mutants. These were: (i) the $\Delta flv1/\Delta flv3$ mutant, deficient in both Flv1 and Flv3 proteins (*Allahverdiyeva et al., 2011*); (ii) $\Delta flv2$ which does not express the Flv2 protein but retains a low amount of Flv4 and WT levels of Sll0218 (*Zhang et al., 2012*); (iii) $\Delta flv4$ which is deficient in the accumulation of all three *flv4-2* operon proteins (*Zhang et al., 2012*); (iv) $\Delta sll0218$ which lacks the small Sll0218 protein, but expresses the Flv2 and Flv4 proteins (*Bersanini et al., 2017*); (v) $\Delta flv3/\Delta flv4$ which is deficient in all four FDPs, whereby the absence of Flv3 results in a strong decrease in Flv1 (*Mustila et al., 2016*) and the inactivation of $\Delta flv4$ affects the expression of the whole *flv4-2* operon (*Zhang et al., 2012*); and, finally (vi) the *flv4-2* operon overexpression strain, *flv4-2/*OE, expressing high amounts of Flv2, Flv4 and Sll0218 (*Bersanini et al., 2014*).

Here, we provide in vivo evidence for Flv2/Flv4 mediated $O_2$ photoreduction in one of the most frequently studied cyanobacterial model organisms, *Synechocystis*. Unlike the powerful and rapid response proteins, Flv1 and Flv3, the Flv2 and Flv4 proteins are dispensable for survival under fluctuating light intensities. The expression of *flv4* and *flv2* under LC was found to be regulated by the pH of the growth media, with significant downregulation observed under strongly alkaline pH conditions. Results from this study provide important insights into the response of photosynthetic organisms to changes in $C_i$ and how they regulate the availability of electron sinks.

## Results

### Extent and kinetics of the Mehler-like reaction in cells acclimated to low (LC) and high $C_i$ (HC) conditions

Application of membrane inlet mass spectrometry (MIMS) with $^{18}O$-enriched oxygen allows differentiation between photosynthetic gross $O_2$ production and $O_2$ uptake under illumination. The *flv4-2/*OE cells, accumulating high amounts of Flv2, Sll0218 and Flv4 both in LC and HC (>1% $CO_2$ in air, HC) conditions (*Bersanini et al., 2014*), demonstrated substantially higher $O_2$ photoreduction rates compared to respective WT cells (*Figure 1A, B and D*). The Flv3 protein level was similar in *flv4-2/*OE and wild-type (WT) cells grown under both LC and HC (*Figure 1C*), strongly supporting the in vivo contribution of *flv4-2* operon proteins to $O_2$ photoreduction during illumination. Gross $O_2$

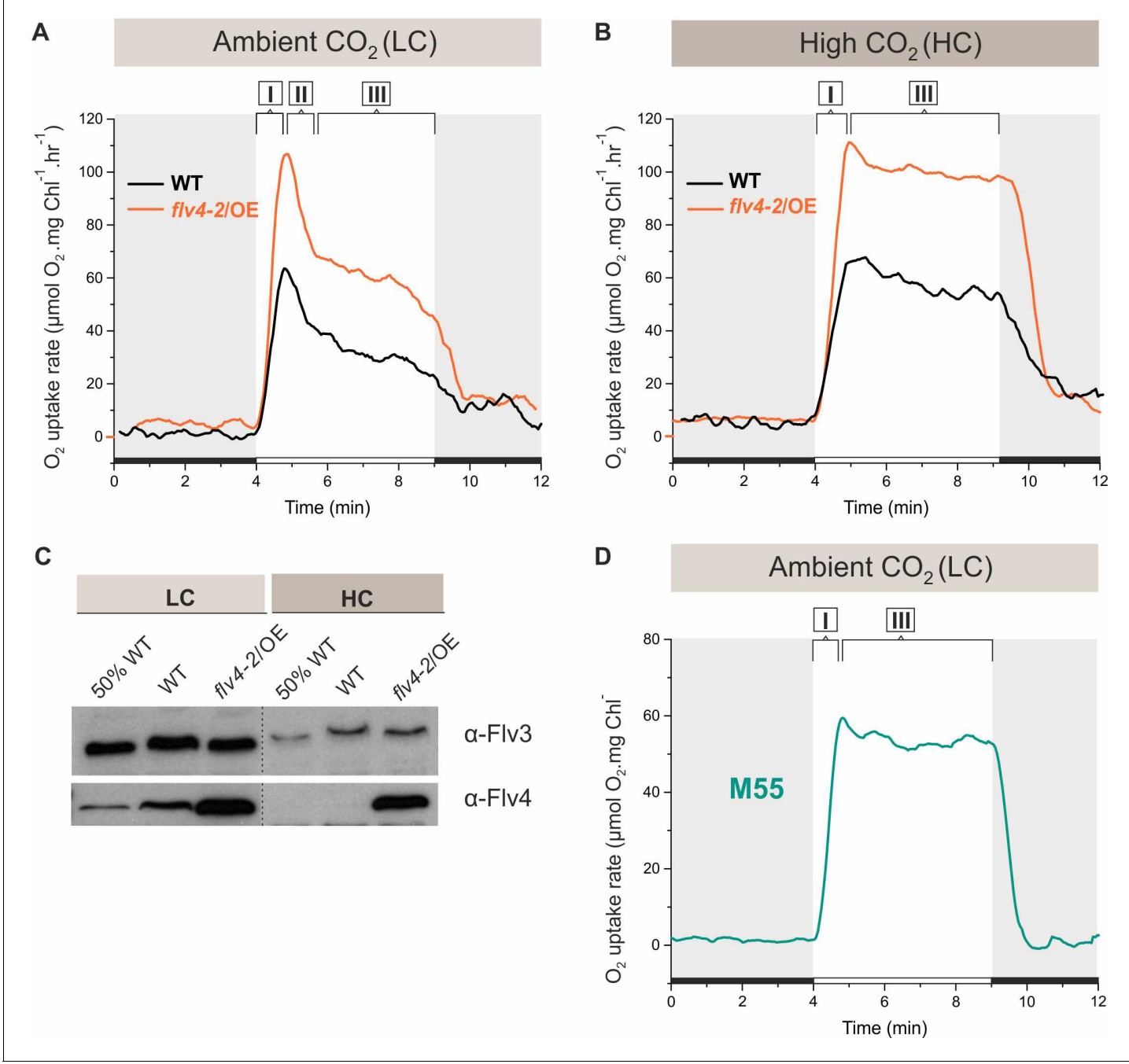

**Figure 1.** $O_2$ reduction rates and Flv3 and Flv4 protein accumulation in cells grown in low (LC) and high $CO_2$ (HC). (A, B) $O_2$ reduction rate of WT, *flv4-2/*OE and (D) the M55 mutant ($\Delta ndhB$) was recorded in darkness (gray background) and under illumination (white background). The experiment was conducted in three independent biological replicates and a representative plot is shown. (*Figure 1—source data 1*). (C) Immunoblot detection of Flv3 and Flv4 in WT and *flv4-2/*OE. Pre-cultures were grown in BG-11, pH 8.2 under 3% $CO_2$ (HC) for 3 days, after that cells were harvested and resuspended in fresh BG-11, pH 8.2 at $OD_{750}$ = 0.2. The experimental cultures were grown under HC or under LC. For the MIMS experiments the cells were harvested and resuspended in fresh BG-11, pH 8.2 at 10 µg Chl *a* $mL^{-1}$. $O_2$ photoreduction was recorded during the transition from darkness to high-light intensity of 500 µmol photons $m^{-2}s^{-1}$. In order to create comparable conditions for MIMS measurements, LC-grown cells were supplemented with 1.5 mM $NaHCO_3$ prior to the measurements. Independent experiments performed on WT cells grown in BG-11 lacking $Na_2CO_3$, but supplied with 1.5 mM $NaHCO_3$ prior to MIMS measurement showed no significant difference in $O_2$ photoreduction rates (*Figure 1—figure supplement 2*), thus allowing confident comparison of the MIMS results. Different phases of $O_2$ photoreduction kinetics are indicated as {I}, {II}, {III}. 50% WT, corresponds to 1:2 diluted WT total protein sample.

DOI: https://doi.org/10.7554/eLife.45766.002

*Figure 1 continued*

The following source data and figure supplements are available for figure 1:

**Source data 1.** $O_2$ reduction rates of WT, flv4-2/OE and M55 mutants grown in different $CO_2$ levels.
DOI: https://doi.org/10.7554/eLife.45766.005
**Source data 2.** Oxygen exchange rates of WT and mutant cells.
DOI: https://doi.org/10.7554/eLife.45766.006
**Figure supplement 1.** $O_2$ reduction rates under high $CO_2$.
DOI: https://doi.org/10.7554/eLife.45766.003
**Figure supplement 2.** $O_2$ reduction rates during the dark-to-light transition of WT cells with and without addition of 1.5 mM $NaHCO_3$ prior MIMS measurements.
DOI: https://doi.org/10.7554/eLife.45766.004

evolution rates of *flv4-2/*OE and WT cells grown under LC did not differ significantly from each other. However, a significant increase in the gross $O_2$ evolution rate was observed in HC grown *flv4-2/*OE cells (*Figure 1—source data 2*).

As reported earlier, the $C_i$ level has a remarkable effect on the expression of FDPs at both transcript and protein level: Flv2, Flv4 and Flv3 have been shown to be strongly upregulated under LC (*Zhang et al., 2009*; *Wang et al., 2004*; *Battchikova et al., 2010*), and down-regulated upon a shift to HC (*Zhang et al., 2009*; *Hackenberg et al., 2009*; *Figure 1C*). Nevertheless, a direct comparison of the efficiency and kinetics of the Mehler-like reaction in HC- and LC-acclimated cells has not been reported, thus the contribution of different FDPs to $O_2$ photoreduction has been difficult to assess. Our initial approach to evaluating the contributions of the different FDPs was based on determining the activity of the Mehler-like reaction in *Synechocystis* cells grown under LC and HC (3% $CO_2$) conditions, at pH 8.2.

After a shift from darkness, WT cells demonstrated a rapid light-induced $O_2$ uptake under both LC and HC conditions (59 ± 6.4 and 56 ± 6.4 µmol $O_2$ mg Chl $a^{-1}$ $h^{-1}$, respectively). This fast induction phase is designated as {I} in *Figure 1A and B*. Yet, the kinetics of $O_2$ photoreduction in LC-grown cells differed from those grown under HC. In the LC-grown WT cells, the fast induction phase {I} was followed by a clear biphasic quenching of $O_2$ reduction, namely by the strong decay phase {II}, which continued for about one minute, followed by a quasi-stable state, phase {III} (~33 ± 5.9 µmol $O_2$ mg Chl $a^{-1}$ $h^{-1}$) during illumination. Contrasting this, in HC-grown WT cell, the light-induced $O_2$ reduction rate achieved in phase {I} declined only slightly during the first 2–3 min (from ~56 ± 7.7 to~48 ± 6.3 µmol $O_2$ mg Chl $a^{-1}$ $h^{-1}$). Thereafter, the rate remained relatively steady for at least 5 min (*Figure 1B*) of illumination. In *flv4-2/*OE cells, grown both in LC- and HC, light-induced $O_2$ reduction was stronger that in the WT. Nevertheless, the kinetic phases of $O_2$ photoreduction in *flv4-2/*OE cells resembled those of respective WT cells, being relatively stable under HC and demonstrating a strong biphasic quenching under LC.

Upon a shift from darkness to light, the Δ*flv2* and Δ*flv4* mutants grown under HC conditions demonstrated a similar $O_2$ photoreduction pattern as the WT (*Figure 1—figure supplement 1*). A negligible amount of Flv2 and Flv4 protein in the WT cells grown under HC (*Zhang et al., 2009*; *Zhang et al., 2012*; *Figure 1C*) explains their lack of contribution to the Mehler-like reaction. The near absence of any light-induced $O_2$ reduction in the Δ*flv3*/Δ*flv4* and Δ*flv1*/Δ*flv3* mutants (*Figure 1—figure supplement 1*) confirms that the small amount of the Flv1/Flv3 heterodimers (decreased Flv3 protein accumulation in HC compared to LC conditions, *Figure 1C*), is responsible for the constant Mehler-like reaction under the HC condition (*Helman et al., 2003*).

To uncover the reason for the fast decay of $O_2$ photoreduction observed under LC conditions (*Figure 1A*), we first tested putative competition between the NAD(P)H:quinone oxidoreductase (NDH-1) complex and FDPs for available photosynthetic electrons. The NDH-1 complex is a powerful machinery utilizing electrons for cyclic electron transport (CET) around PSI, $CO_2$ uptake and respiration under LC conditions (*Zhang et al., 2004*; *Schuller et al., 2019*). To this end, $O_2$ photoreduction was measured in the M55 mutant (Δ*ndhB*), which is deficient in the hydrophobic NdhB subunit (*Ogawa, 1991*) and thus lacks all NDH-1 complexes (*Zhang et al., 2004*). The M55 mutant cells (grown under LC, pH 8.2 conditions) demonstrated a fast induction of $O_2$ photoreduction (phase I) similar to the WT, which continued at steady-state, lacking the second phase of $O_2$ photoreduction after the dark-to-light transition (*Figure 1D*). Importantly, the M55 mutant showed a slow induction

(see phase I of gross $O_2$ evolution in *Figure 1—source data 2*) and considerably lower gross $O_2$ evolution rate compared to the WT cells (see phase III of gross $O_2$ evolution in *Figure 1—source data 2*). This suggests that a steady-state $O_2$ photoreduction in M55 is not due to increased electron flow from PSII. The lack of a strong second phase in $O_2$ photoreduction kinetics resembles the situation in WT cells grown under HC (*Figure 1B*; *Figure 1—figure supplement 1*), where the expression of the NDH-1 complex is strongly reduced, and thus suggests competition for electrons between the NDH-1 complexes and FDPs under LC conditions.

## The extent and kinetics of the Mehler-like reaction are strongly dependent on the pH and carbonate concentration of the growth medium

The pH and the presence of carbonate in the growth medium were evaluated as possible modulators of the extent and kinetics of the Mehler-like reaction and the accumulation of FDPs under LC conditions. Standard BG-11 medium containing sodium carbonate ($Na_2CO_3$) at a final concentration of 0.189 mM was used for all growth experiments, other than those indicated to be $C_i$ limited. In these experiments, performed under atmospheric $CO_2$, $C_i$ limitation was achieved by omitting $Na_2CO_3$ from the BG-11 growth media.

### The effect of pH

The WT cells grown at pH 9 demonstrated a strong but only transient Mehler-like reaction: the $O_2$ photoreduction rate reached its maximum during the first 30 s of illumination, then quickly dropped (within 1 min) to the initial level of dark $O_2$ uptake (*Figure 2*, right panel). Similarly to the WT, the $\Delta flv4$ mutant cells demonstrated only a transient $O_2$ photoreduction upon illumination. There was no significant $O_2$ photoreduction detected for $\Delta flv1/\Delta flv3$ and $\Delta flv3/\Delta flv4$ mutants grown at pH 9.

Immunoblotting using specific antibodies showed that, as for WT cells grown under HC (*Figure 1D*), Flv2 and Flv4 proteins were almost undetectable in the WT grown under LC at pH 9 (*Figure 3A*).

In line with protein data, the transcript levels of both *flv2* and *flv4* were significantly down-regulated in the cells grown at pH 9 (*Figure 3B*), suggesting a pH-dependent transcriptional regulation of *flv4* and *flv2*. This is consistent with earlier transcriptional profiling experiments reporting downregulation of *flv2* and *flv4* transcripts after transferring *Synechocystis* from pH 7.5 to pH 10 (*Summerfield and Sherman, 2008*). Importantly, the accumulation of Flv3 was not affected at pH 9. These results strongly suggest that the conspicuous but transient $O_2$ photoreduction observed in the WT and $\Delta flv4$ mutant cells at pH 9 originates mainly from the activity of Flv1/Flv3 heterodimer.

The WT cells grown at pH 6, at pH 7.5 (*Figure 2*, left and middle panels, respectively) and at pH 8.2 (*Figure 1A*) demonstrated a rapid induction of $O_2$ reduction (phase {I}) followed by a biphasic decay during illumination: a fast decay phase (phase {II}) and a quasi-stable phase (phase {III}) (*Figures 1A* and *2*). The highest $O_2$ photoreduction rate was observed in the WT cells grown at pH 6 (*Figure 2*).

Importantly, the $\Delta flv1/\Delta flv3$ mutant also showed residual $O_2$ photoreduction: only a small $O_2$ uptake was noticeable at pH 7.5, whereas at pH 6 the $O_2$ photoreduction rate was substantial and constant during 5 min of illumination (*Figure 2*). Unlike the $\Delta flv1/\Delta flv3$ mutant, both the $\Delta flv2$ (*Figure 2—figure supplement 1*) and $\Delta flv4$ (*Figure 2*) mutants showed a strong transient $O_2$ photoreduction phase, peaking around the first 30 s of illumination and decaying quickly thereafter. This occurred at all tested pH levels. These results together with those demonstrating highly increased rates of $O_2$ photoreduction in the overexpression strain *flv4-2*/OE (*Figure 1B*) collectively confirm the in vivo involvement of both Flv2 and Flv4 proteins in $O_2$ photoreduction. The $O_2$ photoreduction kinetics of the $\Delta sll0218$ mutant resembled that of the WT (*Figure 1—figure supplement 1* and *Figure 2—figure supplement 1*), indicating that the Sll0218 protein does not contribute to the Mehler-like reaction under the HC and LC conditions studied here. These results led us to exclude the $\Delta sll0218$ mutant from any further experiments included in this section.

The data presented above allowed us to make preliminary conclusions about the origin of the different kinetic phases of $O_2$ photoreduction. Since a transient $O_2$ photoreduction was characteristic for the WT, $\Delta flv2$ and $\Delta flv4$ cells, but almost undetectable for $\Delta flv1/\Delta flv3$, it is conceivable that the Flv1/Flv3 heterodimer is mostly responsible for the strong and transient $O_2$ uptake during dark-light

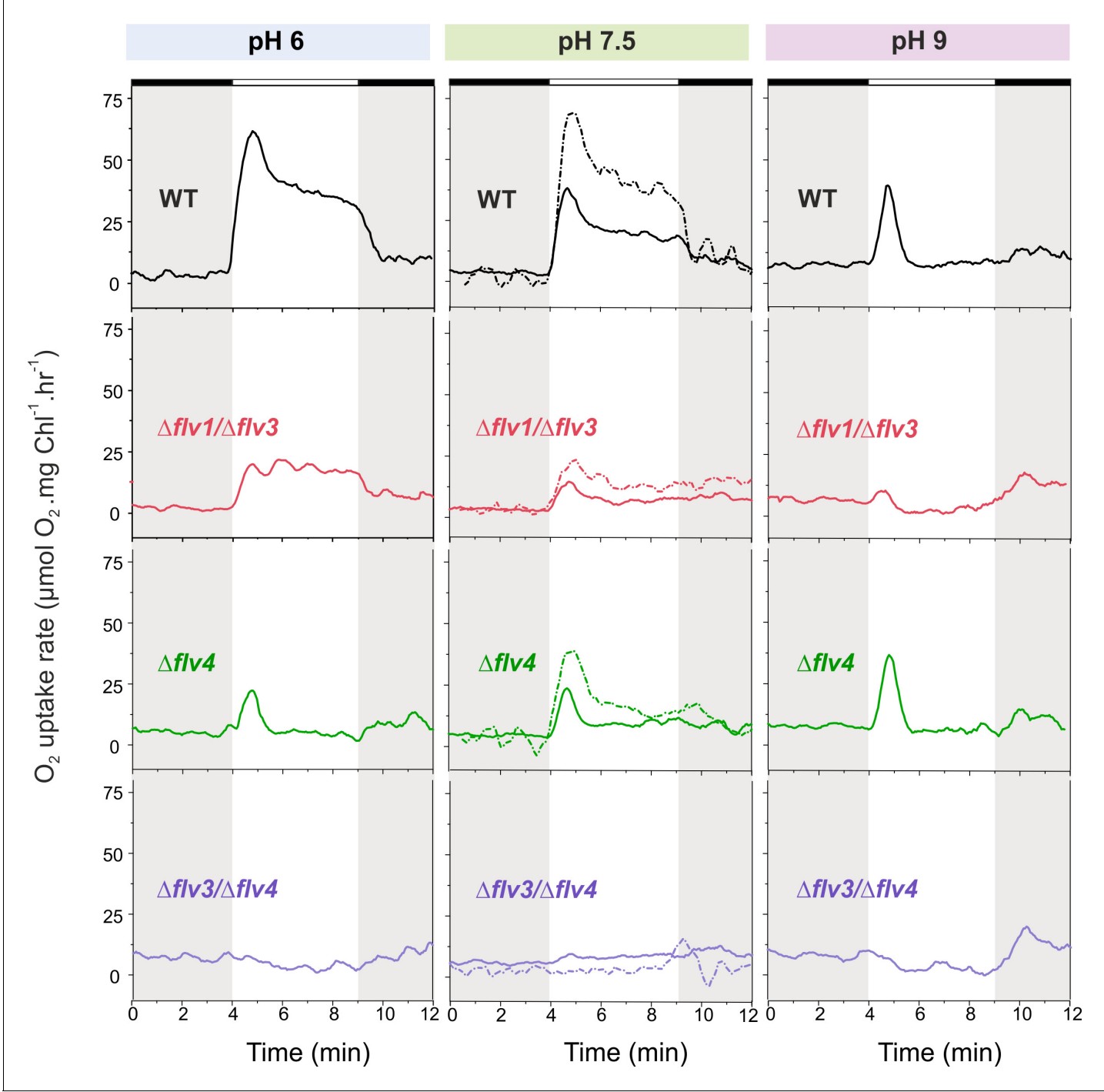

**Figure 2.** $O_2$ reduction rates of WT and FDP mutants grown at different pH levels. $O_2$ reduction rate was recorded in darkness (gray background) and under illumination with actinic white light at an intensity of 500 $\mu$mol photons m$^{-2}$ s$^{-1}$ (white background). Pre-cultures were grown in standard BG-11 medium (containing $Na_2CO_3$ at a final concentration of 0.189 mM) under HC for 3 days at different pH levels. For MIMS experiments, cells were shifted to LC at $OD_{750} \approx 0.2$ (same pH) and grown for 4 days before measurements. Exceptions were: (i) pH 6 experimental cultures were inoculated from pH 8.2 pre-cultures; and (ii) pH 7.5 pre-culture was shifted to LC in standard BG-11 containing $Na_2CO_3$ at a final concentration of 0.189 mM or in BG-11 without $Na_2CO_3$ (dotted line '- $Na_2CO_3$'). The experiment was conducted in three independent biological replicates (except experiment at pH 6 with n = 2 independent biological replicates) and a representative plot is shown. (*Figure 2—source data 1*). In order to create comparable conditions for MIMS measurements, all cells were supplemented with 1.5 mM $NaHCO_3$ prior to the measurements.
DOI: https://doi.org/10.7554/eLife.45766.007

The following source data and figure supplement are available for figure 2:

*Figure 2 continued on next page*

*Figure 2 continued*

**Source data 1.** $O_2$ reduction rates of WT and FDP mutants grown at different pH levels.
DOI: https://doi.org/10.7554/eLife.45766.009
**Figure supplement 1.** $O_2$ photoreduction rates of the $\Delta flv2$ and $\Delta sll0218$ mutants grown at LC pH 7.5 and 8.2 with and without $Na_2CO_3$.
DOI: https://doi.org/10.7554/eLife.45766.008

transitions, whilst Flv2/Flv4 contributes to steady-state $O_2$ photoreduction under LC (see $\Delta flv1/\Delta flv3$ particularly at pH 6, *Figure 2*). The complete lack of $O_2$ photoreduction in the $\Delta flv3/\Delta flv4$ mutant (representing deficiency of all four FDPs) is in line with this hypothesis. Importantly, there was no significant difference in the gross $O_2$ evolution rates observed between the wild-type and the FDP mutants (*Figure 1—source data 2*).

It is not only FDPs, but also distinct variants of the NDH-1 complex as well as $HCO_3^-$ transporters (*Zhang et al., 2004*) which are known to respond to $CO_2$ and pH levels of the growth medium. Immunoblotting was performed to evaluate the abundances of NdhD3, representing a low $C_i$-inducible NDH-1MS complex, and SbtA, a high-affinity low $C_i$-inducible $Na^+/HCO_3^-$ transporter, in WT and different mutants under conditions used for the MIMS experiments.

As expected, in WT cells grown at pH 7.5, NdhD3 and SbtA were not detected under HC conditions, but both proteins were strongly accumulated in LC (*Figure 3D*). However, in LC conditions, the increase in alkalinity of the growth medium to pH 9 resulted in markedly lower levels of NdhD3 and SbtA accumulation compared to those observed at pH 7.5. The effect was more pronounced in the case of SbtA. Interestingly, the $\Delta flv2$ and $\Delta flv4$ mutants demonstrated a decrease of SbtA accumulation compared to WT even at pH 7.5 in LC, whereas in *flv4/OE* SbtA remained at the same level as in WT (*Figure 3D*).

The expression of the SbtA protein closely followed the changes in the expression of Flv2 and Flv4 proteins under all growth conditions, suggesting that Flv2/Flv4 and the $C_i$ uptake mechanisms, particularly the inducible high-affinity $Na^+/HCO_3^-$ transporter, share a common regulatory pathway of protein expression.

Unlike the growth media at pH 6–8.2, the $C_i$-pool at pH 9 contains an additional species, $CO_3^{2-}$. It is possible that a small amount of $CO_3^{2-}$ in the external growth medium acts as a signal to trigger the regulation of *flv2* and *flv4* expression via antisense RNA *as1-flv4* and the master transcription factors, *ndhR* or *cmpR* (*Eisenhut et al., 2012*). Considering that the double negative charge of $CO_3^{2-}$ prevents its diffusion through the cell membrane, and the fact that an active carbonate uptake transporter is currently unknown, we cannot yet consider $CO_3^{2-}$ to be an internal sensor. To gain further insight to the carbonate effect on $O_2$ photoreduction, MIMS experiments were performed on FDP mutants grown in BG-11 medium in the presence (0.189 mM) and absence of sodium carbonate.

## The effect of sodium carbonate

Culturing the cells without $Na_2CO_3$ at pH 7.5 clearly enhanced $O_2$ photoreduction in the WT and all studied FDP mutants (*Figure 2*, middle panel). Despite such a clear variation in $O_2$ photoreduction rates in the WT, no significant difference in gene transcript (*Figure 3B*) and protein levels (*Figure 3C*) of FDPs were observed in the presence or absence of $Na_2CO_3$.

## FDP induced $O_2$ photoreduction does not occur at PSII or PQ-pool level

In order to establish where in the electron transport chain the Flv2/Flv4 heterodimer-related $O_2$ photoreduction occurs, we focused on the *flv4-2/OE* mutant (grown at LC, pH 7.5, without carbonate). This mutant showed especially high accumulation of Flv2 and Flv4 proteins and a higher $O_2$ photoreduction rate than the WT (*Figure 1*). When linear electron transport was blocked at Cytochrome $b_6f$ (Cyt $b_6f$) level using DBMIB as an inhibitor (*Draber et al., 1970*; *Yan et al., 2006*), both the WT (*Ermakova et al., 2016*) and *flv4-2/OE* mutant cells demonstrated a strong light-induced $O_2$ uptake (*Figure 3—figure supplement 1*). As expected, in the $\Delta cyd$ mutant the light-induced $O_2$ uptake was not detected in the presence of DBMIB (*Ermakova et al., 2016*, *Figure 3—figure supplement 1*). The addition of HQNO, an inhibitor of Cytochrome *bd* quinol oxidase (Cyd) (*Pils et al., 1997*) and Cyt $b_6f$ (*Fernández-Velasco et al., 2001*) to the DBMIB-treated WT and *flv4-2/OE* completely

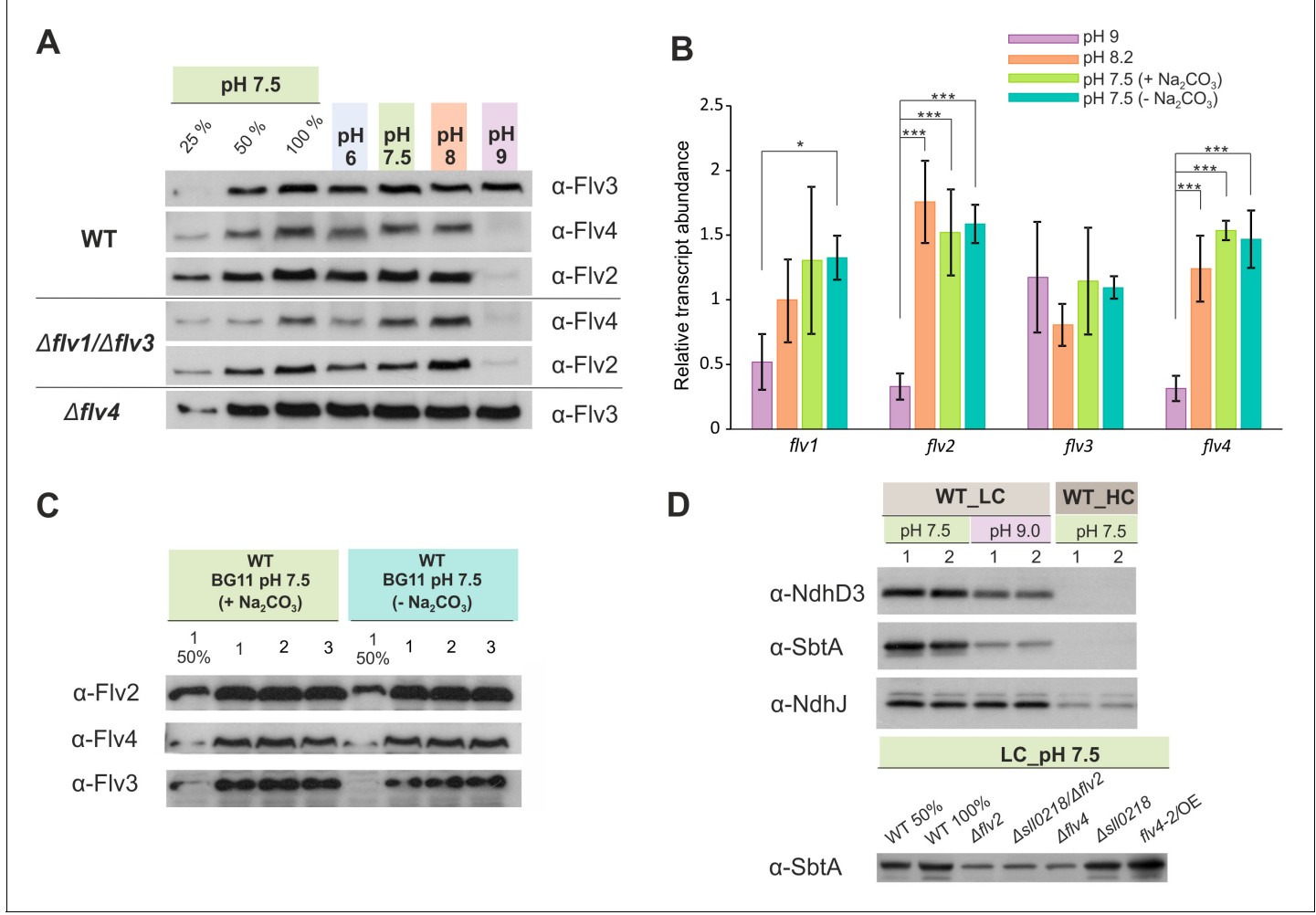

**Figure 3.** The effect of the pH of growth medium on the protein and transcript accumulation. (A, B) The effect of the pH and (B, C) sodium carbonate in the growth medium (A, C) on the protein and (B) transcript levels of FDP. (D) Protein immunoblots demonstrating the accumulation of bicarbonate transporter (SbtA) and NDH-1 subunits (NdhD3 and NdhJ) in the cells grown at different pH and $CO_2$ concentration. Cells were pre-grown at different pH levels (+$Na_2CO_3$) under HC for 3 days, harvested, resuspended in fresh BG-11 (pH maintained), adjusted to $OD_{750} \approx 0.2$ and shifted to LC for 4 days. At pH 7.5, the cells were grown at LC in the presence (+ $Na_2CO_3$, at final concentration of 0.189 mM) or in the absence (- $Na_2CO_3$) of sodium carbonate (B, C). Transcript abundance is presented as mean ± SD, n = 2–4 biological replicates, asterisks indicate a statistically significant difference to the WT (*p<0.05; ***p<0.001) (*Figure 3—source data 1*). Numbers 1–3 indicate different biological replicates. 25% and 50% correspond to 1:4, 1:2 diluted total protein sample, and 100% indicates undiluted total protein sample.

DOI: https://doi.org/10.7554/eLife.45766.010

The following source data and figure supplement are available for figure 3:

**Source data 1.** Transcript abundance of *flv1*, *flv2*, *flv3* and *flv4* genes.
DOI: https://doi.org/10.7554/eLife.45766.012

**Figure supplement 1.** $O_2$ uptake in the WT, *flv4-2*/OE, *Δflv4* and *Δcyd* mutant.
DOI: https://doi.org/10.7554/eLife.45766.011

eliminated $O_2$ photoreduction. These results confirmed that Cyd was solely responsible for the observed $O_2$ photoreduction occurring at the PQ-pool level.

## Growth phenotype of FDP deletion mutants under fluctuating light intensities

We have previously demonstrated that the Flv1/Flv3 heterodimer enables cell growth under fluctuating light, by functioning in the Mehler-like reaction as an efficient electron sink (*Allahverdiyeva et al., 2013*). However, the results of the current study clearly suggest an additional

involvement of the Flv2/Flv4 heterodimer in the Mehler-like reaction, particularly under conditions of LC and at pH values of 8.2 or lower (*Figures 1* and *2*). These findings led us to more precisely examine the combined effects of the pH of the growth medium and the fluctuating growth light conditions (FL) on the growth performance of various FDP mutants. To this end, both severe (FL20/500, when 20 µmol photons $m^{-2} s^{-1}$ background light was interrupted every 5 min by 30 s light pulse intensity of 500 µmol photons $m^{-2} s^{-1}$) and mild (FL50/500, when 50 µmol photons $m^{-2} s^{-1}$ background light was interrupted every 5 min by 30 s light pulse intensity of 500 µmol photons $m^{-2} s^{-1}$) fluctuating lights were applied at different levels of pH. In line with our previous work, the $\Delta flv1/\Delta flv3$ mutant (also $\Delta flv3/\Delta flv4$) failed to grow under severe (FL20/500) light fluctuations, independent of the pH of the growth medium (*Figure 4*; *Figure 4—figure supplement 1*). Differently to the severe FL20/500 condition, under mild fluctuating light (FL50/500), the $\Delta flv1/\Delta flv3$ mutant demonstrated slower growth than the WT under alkaline pH (pH 9, *Figure 4* and pH 8.2 (*Mustila et al., 2016*), *Figure 4—figure supplement 1*)). Growth was similar to the WT at pH 7.5 (*Mustila et al., 2016*), *Figure 4—figure supplement 1*) and pH 6 (*Figure 4*). Importantly, the $\Delta flv4$ mutant grew similarly to the WT at all studied pH levels, both under mild and severe FL conditions (*Figure 4*). The $\Delta flv2$, $\Delta sll0218$ and $flv4-2$/OE mutants also demonstrated similar growth to the WT under severe FL20/500 at pH 7.5 and 8.2 (*Figure 4—figure supplement 1*).

The results above strongly suggest that, in contrast to the Flv1/Flv3-originated Mehler-like reaction, Flv2/Flv4-driven $O_2$ photoreduction is not essential for the survival of cells under fluctuating light.

## Effect of increasing light intensities on the Mehler-like reaction

In order to assess the response of the $O_2$ photoreduction to different light intensities, the WT, $\Delta flv4$ and $\Delta flv1/\Delta flv3$ mutant cells were illuminated with 500, 1000 and 1500 µmol photons $m^{-2} s^{-1}$ white light (*Figure 5*). Under LC conditions, increasing the light intensity from 500 to 1000 µmol photons $m^{-2} s^{-1}$ resulted in a two-fold increase of the maximum $O_2$ photoreduction rate in the WT (*Figure 5A and D*). The further increase (1500 µmol photons $m^{-2} s^{-1}$) only slightly enhanced (2.3-fold) the maximum $O_2$ photoreduction rate, suggesting that the applied light intensity was nearly saturating. Likewise, the $\Delta flv4$ mutant demonstrated about 1.9- and 2.3-fold enhancements of the maximum rate of transient light-induced $O_2$ reduction under 1000 and 1500 µmol photons $m^{-2} s^{-1}$, respectively (*Figure 5C and D*). Contrasting this was the results of the $\Delta flv1/\Delta flv3$ mutant, which showed lesser responses to increasing light intensities (1.6- and 1.8-fold enhancement in the maximum rate at 1000 and 1500 µmol photons $m^{-2} s^{-1}$, respectively) (*Figure 5B and D*). It is important to note that both the $\Delta flv4$ and $\Delta flv1/\Delta flv3$ mutants accumulate nearly the WT level of the Flv3 or Flv4/Flv2 proteins, respectively (*Zhang et al., 2009*; *Mustila et al., 2016*). Moreover, increasing light intensity from 500 to 1500 µmol photons $m^{-2} s^{-1}$ also resulted in enhancement of the $O_2$ photoreduction rate in the WT cells grown under HC (*Figure 5—figure supplement 1*).

The fast and transient response of $\Delta flv4$ mutant cells to drastic increases in light intensity (*Figure 5C*) confirmed the high capacity of Flv1/Flv3-related $O_2$ photoreduction to act as an electron sink. These results explain the essential role of Flv1/Flv3, unlike Flv2/Flv4, for the survival of cells under fluctuating light intensities. Intriguingly, both the fast induction phase {I} and quasi-stable phase {III} of $O_2$ photoreduction rates of the WT were greater than the sum of the individual $O_2$ photoreduction rates from $\Delta flv1/\Delta flv3$ and $\Delta flv4$, implying a strong enhancement of $O_2$ photoreduction by various oligomer activities in the presence of all four FDPs.

Echoing trends seen in $O_2$ photoreduction rates, gross $O_2$ evolution rates of the WT strongly enhanced with increasing light intensities (1.6- and 1.8-fold increase in 1000 and 1500 µmol photons $m^{-2} s^{-1}$, respectively), whereas the $\Delta flv4$ mutant showed only limited increases of gross $O_2$ evolution rates (1.3- and 1.5-fold in 1000 and 1500 µmol photons $m^{-2} s^{-1}$, respectively), and $\Delta flv1/\Delta flv3$ $O_2$ evolution rates were already at maximum levels under the lowest light intensity of 500 µmol photons $m^{-2} s^{-1}$ (*Figure 1—source data 2*). It is worth mentioning that, neither the $\Delta flv1/\Delta flv3$ nor $\Delta flv4$ mutant achieved a steady-state gross $O_2$ evolution during the 5 min of illumination: $\Delta flv1/\Delta flv3$ demonstrated gradual increase, whereas $\Delta flv4$ showed gradual decrease in gross $O_2$ evolution. Next, PSII ($O_2$ evolving activity monitored in the presence of artificial electron acceptor, DMBQ) and PSI (maximum oxidizable amount of P700, $P_m$) activities were measured in cells grown under moderate light (50 µmol photons $m^{-2} s^{-1}$) and exposed to high light (1500 µmol photons $m^{-2} s^{-1}$) for 2 hr. After 2 hr of high light treatment, $\Delta flv1/\Delta flv3$ showed no significant difference in the maximum

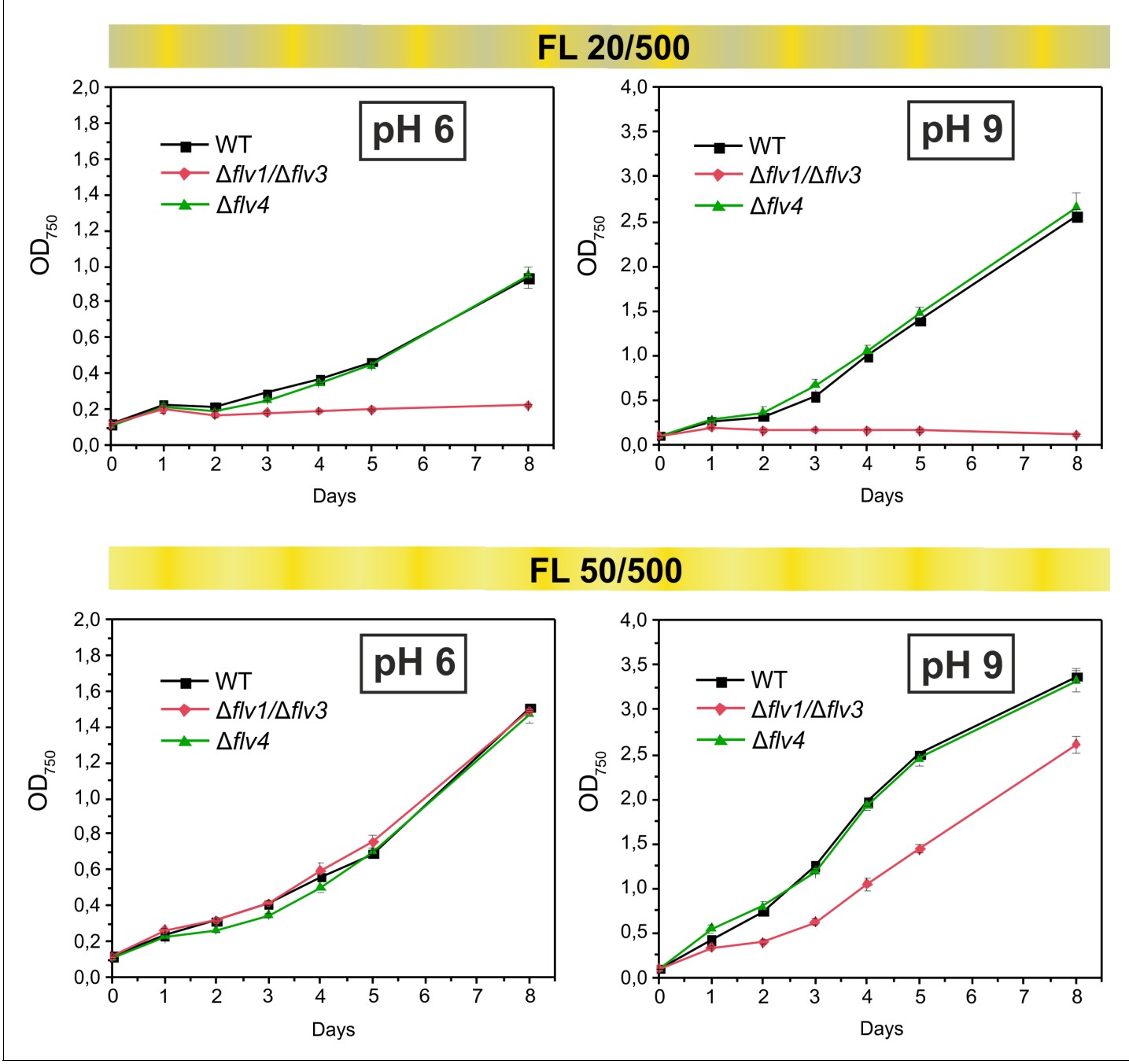

**Figure 4.** Growth curves of the different FDPs mutants under fluctuating light intensities. Pre-cultures were grown in BG-11 medium under HC for 3 days illuminated with constant light of 50 µmol photons $m^{-2} s^{-1}$. The cells pre-grown at pH 9 or pH 8.2 (for experimental culture at pH 6) were harvested, resuspended in fresh BG-11 (pH 9 or 6), adjusted to $OD_{750}$ = 0.1 and shifted to LC. Experimental cultures were grown under FL 20/500 or 50/500 regime for 8 days. The experiment was conducted in two independent biological replicates and average values was plotted.

DOI: https://doi.org/10.7554/eLife.45766.013

The following source data and figure supplement are available for figure 4:

**Source data 1.** Growth of the different FDPs mutants under fluctuating light intensities.

DOI: https://doi.org/10.7554/eLife.45766.015

**Figure supplement 1.** Growth curves of the different FDP mutants under fluctuating light intensities (FL20/500 - 20 µmol photons $m^{-2}s^{-1}$ background light is interrupted with 30 s of 500 µmol photons $m^{-2}s^{-1}$ light every 5 min).

DOI: https://doi.org/10.7554/eLife.45766.014

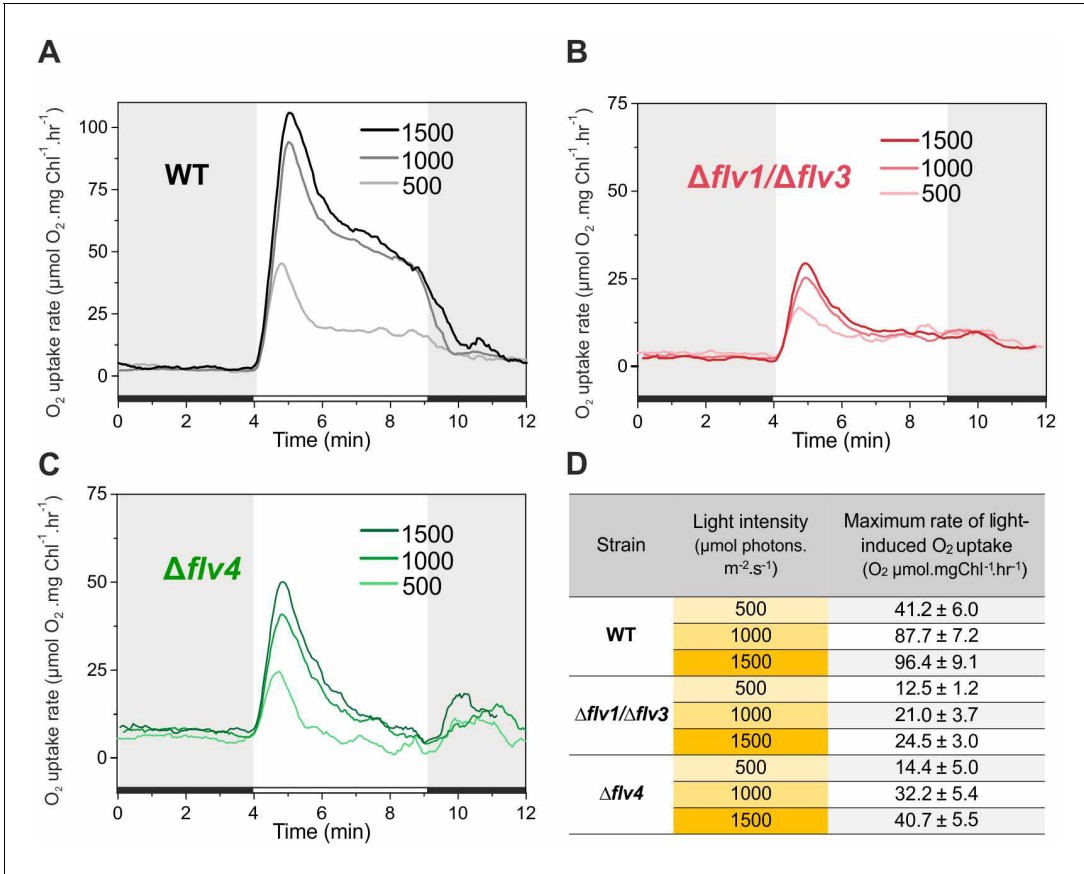

**Figure 5.** Rates of $O_2$ reduction in response to increasing light intensity in WT, $\Delta flv1/\Delta flv3$ and $\Delta flv4$ mutant cells (A, B, C, respectively). $O_2$ reduction rate was recorded in darkness (gray background) and under illumination with actinic white light intensities of 500, 1000 and 1500 μmol photons m$^{-2}$ s$^{-1}$ (white background). In order to create comparable conditions for MIMS measurements, all cells were supplemented with 1.5 mM NaHCO$_3$ prior to the measurements. Pre-cultures were grown in BG-11 medium (pH 7.5) under 3% $CO_2$ (HC) for 3 days and then shifted to LC (atmospheric 0.04% $CO_2$ in air) at OD750 = 0.2 and pH 7.5 for 4 days. For MIMS measurements, cells were harvested and resuspended in fresh BG-11 medium at a Chl $a$ concentration of 10 μg mL$^{-1}$. (D) Maximum rate of light-induced $O_2$ uptake ($O_2$ μmol mgChl $a^{-1}$ hr$^{-1}$) of WT, $\Delta flv1/\Delta flv3$ and $\Delta flv4$ mutant cells at different light intensities applied. The experiment was conducted in three independent biological replicates and a representative plot is shown (**Figure 5—source data 1**).

DOI: https://doi.org/10.7554/eLife.45766.016

The following source data and figure supplements are available for figure 5:

**Source data 1.** Rates of $O_2$ reduction in response to increasing light intensity in WT, $\Delta flv1/\Delta flv3$ and $\Delta flv4$ mutant cells.
DOI: https://doi.org/10.7554/eLife.45766.019

**Figure supplement 1.** Rates of $O_2$ reduction in response to increasing light intensity in WT and $\Delta flv1/\Delta flv3$ mutant cells grown under 3% $CO_2$ (HC).
DOI: https://doi.org/10.7554/eLife.45766.017

**Figure supplement 2.** The maximum oxidisable amount of P700 (Pm) and PSII activity of the WT, $\Delta flv1/\Delta flv3$ and $\Delta flv4$ mutant cells.
DOI: https://doi.org/10.7554/eLife.45766.018

oxidizable amount of P700 ($P_m$) and PSII activity compared to the WT and $\Delta flv4$ mutant (**Figure 5— figure supplement 2**). This is in line with previous studies proving that other photoprotective mechanisms are able to replace Flv1/Flv3 (**Zhang et al., 2009**) unless the cells experience abrupt fluctuations in light intensity (**Allahverdiyeva et al., 2013**). It has already been shown that a strong high light (1500 μmol photons m$^{-2}$ s$^{-1}$) causes slightly slow growth and a short high light treatment decreases PSII activity in the $\Delta flv4$ mutant compared to the WT (**Figure 5—figure supplement 2**; **Zhang et al., 2009**; **Bersanini et al., 2014**; **Bersanini et al., 2017**). Importantly, $\Delta flv4$ demonstrated a $P_m$ level comparable to that of the WT after 2 hr of high-light treatment. This suggests the importance of the Flv2/Flv4 driven steady-state $O_2$ photoreduction in photoacclimation, by the prevention of PSII photodamage caused by the over-reduction of the photosynthetic chain.

## The functional expression of FDPs is highly modulated by $C_i$ conditions and light penetration

The inoculum size (starting $OD_{750}$ value) determines the extent of light penetration upon starting a cultivation. In previous studies, cells were pre-grown in HC, then harvested at late logarithmic phase and inoculated in fresh BG-11 (pH 8.2) at $OD_{750} \approx 0.4$–0.5, before shifting to LC for the next 3 days (*Allahverdiyeva et al., 2011*; *Allahverdiyeva et al., 2013*; *Ermakova et al., 2016*). To ensure better light penetration of the cultures and to improve the acclimation of cells to the conditions used in this study, the experimental WT and $\Delta flv1/\Delta flv3$ cultures were inoculated at a low $OD_{750} \approx 0.1$–0.2 and then cultivated for 4 days (instead of 3 days in previous studies). The WT cells grown under LC from a lower OD ($OD_{750} \approx 0.2$) demonstrated notably higher $O_2$ uptake during illumination, compared to the cells shifted to LC at $OD_{750} \approx 0.5$ (*Figure 6A*). Importantly, the $\Delta flv1/\Delta flv3$ mutant cells shifted to LC at a lower OD ($OD_{750} \approx 0.2$) also demonstrated a residual steady-state $O_2$ photoreduction activity.

Immunoblot analysis using specific FDP antibodies showed that the WT cells transferred from HC to LC at $OD_{750} = 0.2$ accumulated higher amount of the Flv2, Flv3 and Flv4 proteins compared to the cells shifted to LC at $OD_{750} = 0.5$ (*Figure 6B*). A similar trend was also observed in the $\Delta flv1/\Delta flv3$ mutant, which accumulated more Flv2 and Flv4 when cultivated at LC from $OD_{750} = 0.2$. This is in line with previous results showing that the accumulation of *flv2* and *flv4* transcripts in *Synechocystis* (upon a shift from HC to LC, *Zhang et al., 2009*) and vegetative cell-specific *flv1A* and *flv3A*

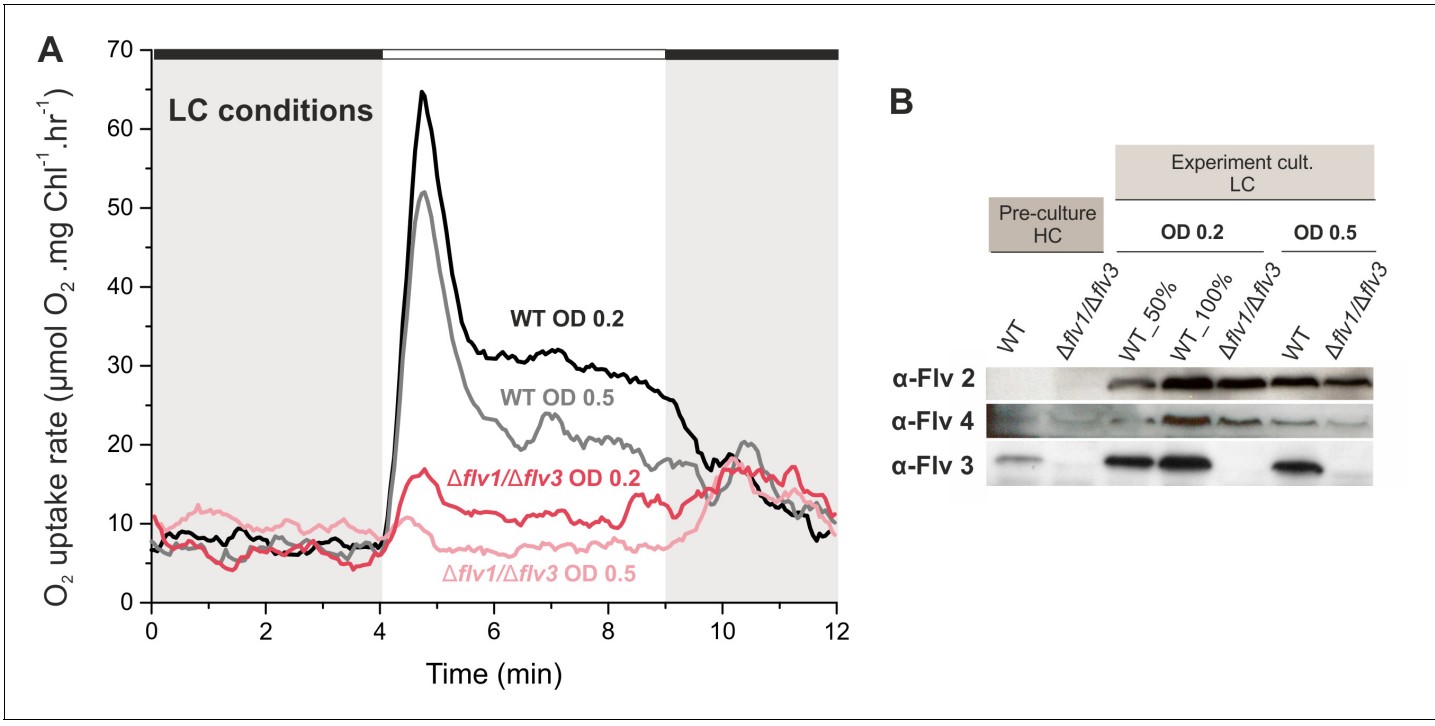

**Figure 6.** Effect of inoculum size on the $O_2$ photoreduction and accumulation of FDPs in the WT and $\Delta flv1/\Delta flv3$ mutant cells. (A) Rates of $O_2$ uptake measured by MIMS during darkness (gray background) and under illumination with actinic white light at an intensity of 500 μmol photos m$^{-2}$s$^{-1}$ (white background). In order to create comparable conditions for MIMS measurements, all cells were supplemented with 1.5 mM NaHCO$_3$ prior to the measurements. (B) Protein immunoblots showing the relative accumulation of different FDPs in the WT and $\Delta flv1/\Delta flv3$ mutant cells. Pre-cultures were grown in BG-11 (pH 8.2) under HC until late logarithmic phase ($OD_{750} \approx 2.5$), then harvested and inoculated in fresh BG-11 under LC at $OD_{750} = 0.2$ for 4 days or $OD_{750} = 0.5$ for 3 days. The experiment was conducted in three independent biological replicates and a representative plot is shown in (A). WT_50% corresponds to 1:2 diluted total protein sample and 100% to undiluted total protein sample.
DOI: https://doi.org/10.7554/eLife.45766.020

The following source data is available for figure 6:

**Source data 1.** Rates of $O_2$ reduction of WT, $\Delta flv1/\Delta flv3$ and $\Delta flv4$ mutant cells grown at different inoculum size.
DOI: https://doi.org/10.7554/eLife.45766.021

transcripts in *Anabaena* sp. PCC 7120 (upon a shift from dark to light, *Ermakova et al., 2013*) strongly depended on light intensity.

The results above highlight that Ci and light penetration upon a shift of cells from pre-culture conditions to different experimental conditions highly modulate the functional expression of FDPs.

## Discussion

### The Flv2/Flv4 heterodimer contributes to the Mehler-like reaction when naturally expressed under LC conditions or artificially overexpressed under HC

By characterizing *Synechocystis* mutants specifically affected in the accumulation of various FDPs, we show here that Flv2 and Flv4, together with Flv1 and Flv3 proteins, are involved in $O_2$ photoreduction in vivo. Until recently, it has generally been accepted that the Flv1/Flv3 proteins safeguard PSI under both HC and LC conditions (*Allahverdiyeva et al., 2013*), whereas proteins encoded by the *flv4-2* operon and being highly expressed under LC, function in the photoprotection of PSII, presumably by directing excess electrons from PSII to an as yet unknown acceptor (*Zhang et al., 2009*; *Zhang et al., 2012*; *Shimakawa et al., 2015*). The possibility of an Flv2/Flv4 contribution to $O_2$ photoreduction in vivo was neglected due to a lack of evidence for light-induced $O_2$ uptake in $\Delta flv1$ and/or $\Delta flv3$ mutants (*Helman et al., 2003*; *Allahverdiyeva et al., 2011*; *Allahverdiyeva et al., 2013*). Thus, Flv1 and Flv3 were assumed to be solely responsible for the Mehler-like reaction. Recently, it was demonstrated that *Synechocystis* Flv4 expressed in *E. coli* is capable of NADH-dependent $O_2$-reduction in vitro (*Shimakawa et al., 2015*). However, the reported reaction rate was extremely low (almost residual) compared to the activity of FDP for example from anaerobic protozoa (*Di Matteo et al., 2008*) and the enzyme showed no affinity to NADPH. A similar scenario was previously presented for the Flv3 protein, where in vitro studies performed on recombinant *Synechocystis* protein led to a claim that Flv3 functions as a homodimer in NADH-dependent $O_2$ reduction (very low affinity to NADPH) (*Vicente et al., 2002*), whilst subsequent study with $\Delta flv1$-OE*flv3* (or $\Delta flv3$-OE*flv1*) mutants clearly demonstrated that homooligomers of Flv3 (or Flv1) do not function in $O_2$ photoreduction in vivo (*Mustila et al., 2016*). Such discrepancies between the in vitro and in vivo results suggest that the in vitro assays conducted thus far have apparently failed to take into full consideration all the complex intracellular interactions, for example the involvement of Fed or FNR as an electron donor for FDPs, or the in vitro experiments do not necessarily demonstrate the processes occurring in vivo.

In this study, we provide compelling evidence for the in vivo contribution of Flv2/Flv4 to $O_2$ photoreduction by applying $^{18}$O-labeled-oxygen and real-time gas-exchange measurements to distinct FDP deletion mutants. The inactivation of *flv2* or *flv4* is shown to result in a substantial decrease of $O_2$ photoreduction in the mutants compared to the WT, while the overexpression of the *flv4-2* operon increases the rate of $O_2$ photoreduction approximately two-fold. In addition, the possibility that the small protein Sll0218 contributes to the Mehler-like reaction is excluded (*Figure 1—figure supplement 1*, compare *Figure 2—figure supplement 1* and *Figure 2*).

It is noteworthy that both the $\Delta flv2$ (deficient in Flv2 but retaining a low amount of Flv4) and $\Delta flv4$ (deficient in both Flv2 and Flv4) mutants showed similar inhibition of $O_2$ photoreduction rates, thus supporting the function of Flv2/Flv4 as a heterodimer in the Mehler-like reaction. The existence of the Flv2/Flv4 heterodimer has been proved biochemically in *Synechocystis* (*Zhang et al., 2012*). Nonetheless, our data do not exclude the possibility that Flv2/Flv2 and/or Flv4/Flv4 homooligomers are also involved in processes other than $O_2$ photoreduction. Such a situation occurs with the Flv1 and Flv3 proteins, which contribute as homooligomers to the photoprotection of cells under fluctuating light conditions, probably *via* an unknown electron transport and/or regulatory network (*Mustila et al., 2016*).

The complete elimination of light-induced $O_2$ reduction in WT cells grown at pH 8.2 (*Ermakova et al., 2016*) or at pH 7.5 (*Figure 3—figure supplement 1*) in the presence of electron-transport inhibitors DBMIB (blocks Qo site of Cyt$b_6f$; *Roberts and Kramer, 2001*) and HQNO (blocks Q$_i$ site of Cyt$b_6f$; *Fernández-Velasco et al., 2001* and also *Pils et al., 1997*) suggests that FDP-driven $O_2$ photoreduction (neither by Flv1/Flv3 nor by Flv2/Flv4) does not occur at the PSII or PQ-pool level. This conclusion is also supported by the fact that, differently to the WT and mutants

deficient in FDPs, the $\Delta cyd$ mutant does not exhibit a light induced $O_2$ uptake in the presence of DBMIB (*Ermakova et al., 2016*; *Figure 3—figure supplement 1*).

From the results discussed above, it can be concluded that both the Flv1/Flv3 and Flv2/Flv4 heterodimers have capacity to drive the Mehler-like reaction, functioning downstream of PSI.

## The Flv1/Flv3 heterodimer drives a strong and steady-state $O_2$ photoreduction under HC

It is generally accepted that under LC conditions, the slowing down of the Calvin-Benson cycle leads to a build-up of reduced stromal components (*Cooley and Vermaas, 2001*; *Holland et al., 2015*), which would stimulate the Mehler reaction to dissipate excess electrons (*Ort and Baker, 2002*). However, under HC conditions, the Mehler reaction would be expected to direct relatively low electron flux to $O_2$. In this study, we provide evidence that HC-grown WT cells are capable of equally high $O_2$ photoreduction as respective LC-grown WT cells, and that cells are capable of maintaining the steady-state activity at least during the first 5–10 min of illumination (*Figure 1A*). Compared to the WT, a drastically lower $O_2$ photoreduction rate is observed in the $\Delta flv1/\Delta flv3$ and $\Delta flv3/\Delta flv4$ mutants grown in HC, confirming that $O_2$ uptake under these conditions is mostly due to the Flv1/Flv3-driven Mehler-like reaction (*Figure 1—figure supplement 1*).

It is important to note that the $O_2$ photoreduction capacity of *Synechocystis* generally correlates with the abundance of FDPs (*Figures 1* and *6*). However, protein abundance is not the only factor that determines $O_2$ photoreduction capacity. Indeed, despite strong and steady-state $O_2$ photoreduction, HC-grown cells demonstrate nearly undetectable levels of Flv2 and Flv4 and low amount of Flv3, compared to levels observed under LC conditions. Furthermore, the increase in $O_2$ photoreduction rates (*Figure 2*, middle panel) obtained by omitting sodium carbonate from the BG-11 growth media at pH 7.5, does not correlate with any significant change in transcript and protein levels of the FDPs, thus suggesting a possible redox regulation of the enzyme activity.

## Under LC, the Flv1/Flv3 heterodimer is a rapid, strong and transient electron sink whereas Flv2/Flv4 supports steady-state $O_2$ photoreduction

The Mehler-like reaction of WT cells grown under LC at pH 6–8.2 exhibits triphasic kinetics of $O_2$ photoreduction originating from the activity of both Flv1/Flv3 and Flv2/Flv4 heterodimers (*Figure 2*). In this study, we were able to unravel the contribution of Flv1/Flv3 and Flv2/Flv4 heterodimers to the $O_2$ photoreduction kinetics: Flv1/Flv3 is mainly responsible for the rapid transient phase, whereas Flv2/Flv4 mostly contributes to the slow steady-state phase.

The almost complete absence of Flv2 and Flv4 proteins in WT cells grown under LC at pH 9 provides an excellent model system, where the Mehler-like reaction is naturally driven solely by the Flv1/Flv3 heterodimer, as is also the case under HC conditions. However, in contrast to HC-grown cells, where Flv1/Flv3 can drive a steady-state $O_2$ photoreduction, the cells grown under LC at pH 9 demonstrate strong but only transient $O_2$ photoreduction, which decays during the first 1–2 min of illumination (*Figure 2*). The identical $O_2$ photoreduction kinetics of the WT cells grown at pH 9 (accumulating Flv3 but lacking both the Flv2 and Flv4 proteins) and the $\Delta flv4$ mutant (accumulating Flv3 but lacking Flv4 and also Flv2), together with the complete absence of $O_2$ photoreduction in the $\Delta flv3/\Delta flv4$ mutant demonstrate that under LC, the Flv1/Flv3 heterodimer contributes to the Mehler-like reaction in a fast and transient manner (*Figure 2*). A similar conclusion was previously suggested for *Synechocystis* (*Allahverdiyeva et al., 2013*) and for the FlvA and FlvB proteins in *Physcomitrella patens* (*Gerotto et al., 2016*) and *Chlamydomonas reinhardtii* (*Chaux et al., 2017*; *Jokel et al., 2018*).

The sole contribution of Flv2/Flv4 to the Mehler-like reaction is clearly demonstrated as a steady-state $O_2$ photoreduction by the $\Delta flv1/\Delta flv3$ mutant grown under LC at pH 6 (*Figure 2*), whilst the same mutant cells grown at pH 7.5 and 8.2 show only residual steady-state $O_2$ photoreduction. It is important to note that the Flv2/Flv4 heterodimer, when expressed, can readily contribute to $O_2$ photoreduction under HC, as demonstrated by the *flv4-2/*OE strain (*Figure 1*), thus excluding all redox and structural hindrances for Flv2/Flv4 to function in $O_2$ photoreduction under HC. However, such a contribution is naturally abolished in WT cells grown under high levels of $CO_2$ by the down-regulation of the *flv4-2* operon (*Zhang et al., 2009*; *Zhang et al., 2012*).

The rate of the Mehler-like reaction in WT cells exceeds the cumulative $O_2$ photoreduction driven solely by Flv1/Flv3 (observed in Δ*flv4*) and Flv2/Flv4 (observed in Δ*flv1*/Δ*flv3*). This demonstrates that all four FDPs are required for an efficient Mehler-like reaction in WT cells upon growth under LC (except at pH 9). A complex interaction between FDPs possibly arises from a coordinated inter-regulation of Flv1/Flv3 and Flv2/Flv4 heterodimers and on the possible occurrence of some active Flv1-4 oligomers (*Figure 7*). Despite detection of homotetrameric organization of *Synechocystis* Flv3 in vitro (*Mustila et al., 2016*), the direct biochemical demonstration of homo- or heterotetramer structures and function in vivo is still missing.

The growth inhibition of Δ*flv1*/Δ*flv3* cells under severe fluctuating light conditions (FL 20/500) at pH 8.2 (*Allahverdiyeva et al., 2013*), pH 7.5 (*Mustila et al., 2016*), pH 6 and pH 9 (*Figure 5*)

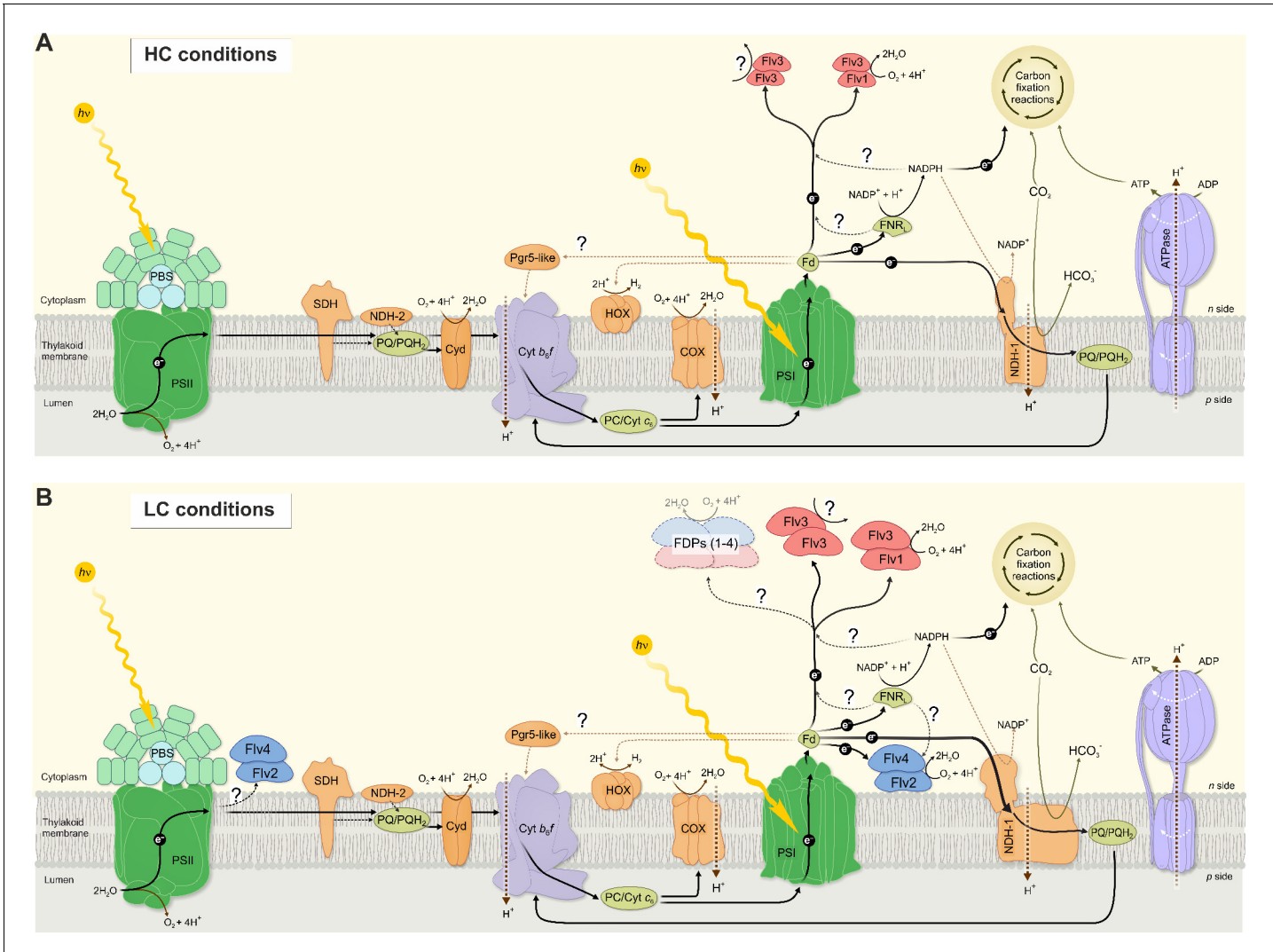

**Figure 7.** A schematic drawing of photosynthetic light reactions and alternative electron transport routes. (A) A steady-state Mehler-like reaction in HC is carried out by the low-abundant, yet catalytically efficient Flv1/Flv3 heterodimer. The Flv3/Flv3 homooligomer is involved in photoprotection as an electron valve with unknown acceptor or as a component of a signaling/regulating network (*Mustila et al., 2016*). (B) In LC-grown cells the two pairs of FDP heterodimers are involved in the Mehler-like reaction: Flv1/Flv3 mainly drives rapid and transient $O_2$ photoreduction and Flv2/Flv4 operates relatively slowly and provides a steady-state background $O_2$ photoreduction. The soluble Flv1/Flv3 heterodimers function as an immediate acceptor of electrons presumable from reduced Fed, whereas association of Flv2/Flv4 with the thylakoid membrane (and/or Flv1/Flv3) is controlled by *pmf* and $Mg^{2+}$. Several oligomeric forms of FDPs are hypothesized to exist, including a heterotetramer comprising different FDP protein compositions. The higher abundance of total NDH-1 complexes and FDPs oligomers in LC conditions, compared to HC conditions, is represented by larger size of the protein complexes.
DOI: https://doi.org/10.7554/eLife.45766.022

demonstrate the essential role of Flv1 and/or Flv3 during drastic changes of light intensity, whereas Flv2 and Flv4 are dispensable under the same conditions (*Figure 4*, *Figure 4—figure supplement 1*). Here, we demonstrate that the crucial importance of Flv1/Flv3 heterodimers is based on their high capacity to rapidly and effectively respond to increasing light intensities (*Figure 5*). By adjusting their $O_2$ photoreduction activity, the Flv1/Flv3 heterodimer works as an efficient and fast sink of electrons, whereas the responsiveness of Flv2/Flv4 is relatively limited and the heterodimer mostly functions on a slow time-scale in steady-state $O_2$ photoreduction.

The intracellular location of these enzymes may partially contribute to the difference in $O_2$ photoreduction: Flv1 and Flv3 are soluble cytosolic proteins able to quickly associate with soluble Fed and direct electrons towards $O_2$ photoreduction. In line with this, the possible interaction between *Synechocystis* Flv1, Flv3 and Fed (*Hanke et al., 2011*), Flv3 and Fed9 (*Cassier-Chauvat and Chauvat, 2014*), *Chlamydomonas reinhardtii* FLVB and FED1 (*Peden et al., 2013*) have been reported. The Flv2/Flv4 heterodimer, specific for cyanobacteria, was suggested to bind to the thylakoid membrane upon increases in $Mg^{2+}$ concentration on the cytoplasmic surface of the thylakoid membrane when lights are turned on (*Zhang et al., 2012*). It is likely that the association of Flv2/Flv4 with the membrane enhances electron transfer from Fed (or FNR) to Flv2/Flv4 and would probably result in a delayed and limited $O_2$ photoreduction activity by Flv2/Flv4. However, the possibility that FDPs accept electrons from different and specific Fed paralogs cannot be excluded.

## Traffic downstream of PSI affects the FDP-mediated Mehler-like reaction

Unlike WT cells demonstrating biphasic decay kinetics of $O_2$ photoreduction under LC conditions (*Figure 1A* and *Figure 2*), the M55 mutant (deficient in NDH-1 mediated CET, $CO_2$ uptake and respiration) (*Ohkawa et al., 2000*) shows steady-state $O_2$ photoreduction, similar to the HC-grown WT (*Figure 1B and D*). This suggests that the strongly upregulated NDH-1 complex under LC in *Synechocystis* (*Zhang et al., 2004*) contributes to a rapid quenching of $O_2$ photoreduction (*Figure 1A*, phase {II}) by efficient withdrawal of electrons from reduced Fed. Under such circumstances, the low but steady-state activity of the Flv2/Flv4 heterodimer is likely to be important for keeping linear electron transport in an oxidized state. This would explain why the PQ-pool is more oxidized in the presence of Flv2/Flv4 and more reduced in its absence, indirectly affecting PSII activity (*Zhang et al., 2012*; *Bersanini et al., 2014*) and *Chukhutsina et al., 2015*). Thus, by allocating different roles for FDPs between the two pairs of heterodimers (Flv1/Flv3 and Flv2/Flv4), the cells are well positioned to respond appropriately to changing $C_i$ levels as well as to abrupt changes in light intensity, in a coordinated and energetically efficient manner.

Unlike prokaryotic cyanobacteria, chlorophytic algae (*e.g. Chlamydomonas reinhardtii*) and mosses rely not only on the FDP-driven pathway, but also harbor the PROTON GRADIENT REGULATION5 (PGR5)/PGR5-LIKE PHOTOSYNTHETIC PHENOTYPE 1 (PGRL1) pathway which operates concomitantly to protect the cells under fluctuating light. It is noteworthy, however, that the PGR5/PGRL1 machinery in *Chlamydomonas reinhardtii* is neither fast nor strong enough to mitigate acceptor-side pressure under highly fluctuating light intensities. To complement this deficiency, the FDP-mediated pathway is indispensable for coping with sudden increases in light intensity (*Jokel et al., 2018*). Interestingly, the introduction of *Physcomitrella patens* FDPs rescues a fluctuating light phenotype of the PGR5 *Arabidopsis thaliana* mutant (*Yamamoto et al., 2016*; *Yamamoto and Shikanai, 2019*), and alleviates PSI photodamage in the PGR5-RNAi, *crr6* (defective in NDH-dependent CET) and the PGR5-RNAi *crr6* double mutants of *Oryza sativa* by acting as a safety valve under fluctuating light and substituting for CET without competing with $CO_2$ fixation under constant light (*Wada et al., 2018*). Moreover, the expression of *Synechocystis* Flv1 and Flv3 in tobacco plants enhances photosynthetic efficiency during dark-light transitions by providing an additional electron sink (*Gómez et al., 2018*). Although data on Flv2/Flv4 proteins expressed in angiosperms is not yet available, our results collectively suggest that the FDP pathway(s) is important to consider in future high-yield crop development and microbial cell factories.

The question of how FDPs avoid competition with $CO_2$ fixation is an interesting one. Relevant mechanisms may include post-transcriptional modifications of the FDPs, such as phosphorylation (*Angeleri et al., 2016*), and/or *pmf* based regulation systems.

*Figure 7* provides a summary scheme of our understanding of the function and interaction of the different FDPs and their oligomers in photoprotection of the photosynthetic apparatus in the model

cyanobacterium *Synechocystis* sp. PCC 6803. The importance of the available $C_i$ species in the function and accumulation of FDPs is emphasized by separate schemes for the HC and LC growth conditions.

# Materials and methods

**Key resources table**

| Reagent type (species) or resource | Designation | Source or reference | Identifiers | Additional information |
|---|---|---|---|---|
| Strain, strain background (*Synechocystis* sp. PCC 6803) | WT, Wild-type | *Williams, 1988* | | |
| Genetic reagent (*Synechocystis* sp. PCC 6803) | Δ*flv2* | *Zhang et al., 2012* | | |
| Genetic reagent (*Synechocystis* sp. PCC 6803) | Δ*flv4* | *Zhang et al., 2012* | | |
| Genetic reagent (*Synechocystis* sp. PCC 6803) | Δ*flv1*/Δ*flv3* | *Allahverdiyeva et al., 2011* | | |
| Genetic reagent (*Synechocystis* sp. PCC 6803) | Δ*flv3*/Δ*flv4* | *Helman et al., 2003* | | |
| Genetic reagent (*Synechocystis* sp. PCC 6803) | Δ*sll0218-flv2* | *Helman et al., 2003* | | |
| Genetic reagent (*Synechocystis* sp. PCC 6803) | *flv4-2*/OE | *Bersanini et al., 2014* | | |
| Genetic reagent (*Synechocystis* sp. PCC 6803) | Δ*sll0218* | *Bersanini et al., 2017* | | |
| Antibody | α-Flv2 (rabbit polyclonal) | AntiProt, against amino acids 521–535 of *Synechocystis* Flv2 | | (1:500) |
| Antibody | α-Flv3 (rabbit polyclonal) | AntiProt, against amino acids 377–391 of *Synechocystis* Flv3 | | (1:2000) |
| Antibody | α-Flv4 (rabbit polyclonal) | AntiProt, against amino acids 412–426 of Synechocystis Flv4 | | (1:500) |
| Antibody | α-NdhD3 (rabbit polyclonal) | Eurogentec, against amino acids 185 to 196 and 346 to 359 of *Synechocystis* NdhD3 | | (1:1000) |
| Antibody | α-SbtA | Kind gift from T. Ogawa, against amino acids 184 to 203 of *Synechocystis* SbtA | | (1:5000) |
| Antibody | α-NdhJ | Kind gift from J. Appel | | (1:1000) |
| Antibody | Secondary antibody, Amersham ECL Rabbit IgG, HRP-linked F(ab')$_2$ fragment (from donkey) | GE Healthcare | NA9340-1ML | (1:10000) |

*Continued on next page*

*Continued*

| Reagent type (species) or resource | Designation | Source or reference | Identifiers | Additional information |
|---|---|---|---|---|
| Commercial assay or kit | Amersham ECL Western Blotting Detection Reagent | GE Healthcare | RPN2209 | |
| Commercial assay or kit | iScript cDNA Synthesis Kit | BioRad, USA | Cat. #170–8891 | |
| Commercial assay or kit | iQ SYBR Green Supermix | BioRad, USA | Cat. #170–8882 | |
| Software, algorithm | qbase + software | Biogazelle, Zwijnaarde, Belgium - www.qbaseplus.com | | |

## Strains and culture conditions

The glucose-tolerant *Synechocystis* sp. PCC 6803 was used as wild type (WT) strain (*Williams, 1988*). The FDP inactivation mutants Δ*flv2*, Δ*flv4* (*Zhang et al., 2012*), and the double mutants Δ*flv1*/Δ*flv3* (*Allahverdiyeva et al., 2011*), and Δ*flv3*/Δ*flv4* (*Helman et al., 2003*), Δ*sll0218*-*flv2* (*Helman et al., 2003*) have been described previously. The *flv4-2*/OE and Δ*sll0218* mutants were described in *Bersanini et al. (2014)*; *Bersanini et al. (2017)*.

Pre-experimental cultures were grown at 30°C in BG-11 medium, illuminated with continuous white light of 50 μmol photons $m^{-2} s^{-1}$ (growth light: GL), under air enriched with 3% $CO_2$ (high carbon: HC). BG-11 medium was buffered with 20 mM 2-(N-morpholino) ethanesulfonic acid (MES, pH 6.0), 20 mM HEPES-NaOH (pH 7.5), 10 mM TES-KOH (pH 8.2) or 10 mM N-Cyclohexyl-2-aminoethanesulfonic acid (CHES, pH 9.0), according to the pH of the experimental condition. Pre-cultures were harvested at logarithmic growth phase, inoculated in fresh BG-11 medium at $OD_{750}$ = 0.2 (or $OD_{750}$ = 0.5 when mentioned), measured with and shifted to low $CO_2$ (atmospheric 0.04% $CO_2$ in air, LC). $OD_{750}$ was measured using Lambda 25 UV/VIS spectrometer (PerkinElmer, USA). HC experimental cultures were inoculated at $OD_{750}$ = 0.1 and kept at HC for 3 days. During experimental cultivation, cells were grown under continuous GL at 30°C with agitation at 120 rpm and without antibiotics. For growth curves, cells pre-cultivated under continuous GL and HC were collected, inoculated at OD750 = 0.1 and shifted to LC under a light regime with a background light of 20 μmol photons $m^{-2} s^{-1}$ interrupted with 500 μmol photons $m^{-2} s^{-1}$ for 30 s every 5 min (FL 20/500) or 50 μmol photons $m^{-2} s^{-1}$ interrupted with 500 μmol photons $m^{-2} s^{-1}$ for 30 s every 5 min (FL 50/500). The standard BG-11 medium used in this work contains sodium carbonate ($Na_2CO_3$) at a final concentration of 0.189 mM and only when mentioned the sodium carbonate was omitted from the growth medium.

Absence of contamination with heterotrophic bacteria was checked by dropping liquid culture on LB and R2A agar plates and kept at 30°C.

## Isolation of total RNA and Real-time quantitative PCR (RT-qPCR)

Total RNA was isolated from exponentially growing *Synechocystis* by hot-phenol method previously described (*Tyystjärvi et al., 2001*). After removing any residual genomic DNA, the RNA concentration and purity were measured with a NanoDrop spectrophotometer (Thermo Scientific, USA). RNA integrity was verified by agarose gel electrophoresis.

Complementary DNA was synthesized from 1 μg of purified RNA using the iScript cDNA Synthesis Kit (BioRad, USA) according to the manufacturer's protocol. Synthesized cDNA was diluted fourfold and used as template for the RT-qPCR. The samples for RT-qPCR were labeled by iQ SYBR Green Supermix (BioRad, USA) to detect accumulation of amplicons in 96-well plates. The primers to detect transcripts of *flv1* and *flv2* as well as for the reference genes *rnpB* and *rimM* are described in *Mustila et al. (2016)*. The forward and reverse primers for *flv3* were 5'-CAACTCAATCCCCGCA TTAC-3' and 5'-CAGTGGAGATTCGGAGCACT-3' and for *flv4* 5'-ACGATGCCTGGAGTCAAAAC-3' and 5'-GGGTATCCGCCACACTTAGA-3'. The PCR protocol was as follows: 3 min initial denaturation of cDNA at 95°C, followed by 40 cycles of 95°C for 10 s, annealing in 57°C for 30 s and extension in

72°C for 35 s. A melting curve analysis was performed at the end. Relative changes in the gene expression were determined using the qbase + software by Biogazelle. One-way ANOVA analysis performed with SigmaPlot was used to determine significant changes in gene expression.

## MIMS experiments

In vivo measurements of $^{16}O_2$ (mass 32) and $^{18}O_2$ (mass 36) exchange was performed using a Membrane-inlet mass spectrometry (MIMS) as described previously in *Mustila et al. (2016)*. Cells were harvested, adjusted to 10 μg Chl *a* mL$^{-1}$ in fresh BG-11 medium and acclimated for 1 hr to the same experimental conditions as was applied for the cultivation.

## Protein isolation, electrophoresis and immunodetection

Total cell extracts and the soluble fractions of *Synechocystis* cells were isolated as described (*Zhang et al., 2009*). Proteins were separated by 12% (w/v) SDS-PAGE containing 6 M urea and transferred onto a PVDF membrane (Immobilon-P; Millipore, Germany) and immunodetected by protein specific antibodies. Horseradish peroxidase (HRP) conjugated secondary antibody (anti-rabbit IgG from donkey) was used for recognizing the primary antibodies and Amersham ECL Western Blotting Detection Reagent (GE Healthcare) was used for the visualization of the antibodies.

## Acknowledgements

The authors would like to thank Dr. Lauri Nikkanen and Dr. Natalia Battchikova for critical reading of the manuscript. Dr. Duncan Fitzpatrick is acknowledged for technical assistance and maintenance of MIMS.

## Additional information

### Funding

| Funder | Grant reference number | Author |
|---|---|---|
| Suomen Akatemia | 315119 | Yagut Allahverdiyeva |
| NordForsk | 82845 | Eva-Mari Aro<br>Yagut Allahverdiyeva |
| Koneen Säätiö | 7fa491 | Yagut Allahverdiyeva |
| Suomen Akatemia | 307335 | Eva-Mari Aro |

The funders had no role in study design, data collection and interpretation, or the decision to submit the work for publication.

### Author contributions

Anita Santana-Sanchez, Conceptualization, Formal analysis, Validation, Investigation, Visualization, Methodology, Writing—original draft, Writing—review and editing; Daniel Solymosi, Conceptualization, Validation, Investigation, Methodology, Writing—original draft, Writing—review and editing; Henna Mustila, Luca Bersanini, Conceptualization, Investigation, Writing—review and editing; Eva-Mari Aro, Conceptualization, Resources, Funding acquisition, Writing—review and editing; Yagut Allahverdiyeva, Conceptualization, Resources, Supervision, Funding acquisition, Validation, Investigation, Visualization, Methodology, Writing—original draft, Project administration, Writing—review and editing

### Author ORCIDs

Anita Santana-Sanchez (ID) https://orcid.org/0000-0002-1556-0321
Yagut Allahverdiyeva (ID) https://orcid.org/0000-0002-9262-1757

### Decision letter and Author response

Decision letter https://doi.org/10.7554/eLife.45766.025
Author response https://doi.org/10.7554/eLife.45766.026

## Additional files

### Supplementary files
• Transparent reporting form
DOI: https://doi.org/10.7554/eLife.45766.023

### Data availability
All data generated or analysed during this study are included in the manuscript and supporting files.

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
