## [Decision Letter]

[Editors’ note: this article was originally rejected after discussions between the reviewers, but the authors were invited to resubmit after an appeal against the decision.]

Thank you for submitting your work entitled "Flv1-4 proteins function in versatile combinations in O_2_ photoreduction in cyanobacteria" for consideration by *eLife*. Your article has been reviewed by three peer reviewers, and the evaluation has been overseen by a Reviewing Editor and a Senior Editor.

The reviewers carefully evaluated your interesting work and concluded that the manuscript is currently too preliminary for publication. They particularly pointed out that data on O_2_ evolution would be necessary to validate your conclusions on O_2_ uptake. It is the policy of *eLife* to avoid lengthy re-review processes with uncertain outcomes. Therefore, we decided to reject the paper at this stage. Please find the specific reviewer comments below.

*Reviewer #1:*

The authors analyze the functions of FLVs to catalyze Mehler reaction in S.6803 using MIMS. The reviewer cannot find any novel thing of FLVs in their physiological functions. Helman et al. (2003) already reported that FLVs catalyze O_2_-dependent electron flow in vivo using MIMS. Furthermore, the authors in the present manuscript already reported the functions of FLV2/4 to catalyze Mehler reaction. As an important, serious issue, the authors did not show any data for 16 O_2_ evolution in PSII, which reflects the electron flux in photosynthetic linear electron flow in vivo. The reviewer cannot understand why the authors did not show these data. The 162-evolution rate greatly affects photosynthetic electron transport rate. Miyake'group already showed the deficiency of FLV suppresses photosynthetic linear electron transport rate, and also suppresses CO_2_-dependent photosynthesis rate (Shaku et al., 2016). The deficiency of FLV induces the reduction of plastoquinone pool, which suppresses Q-cycle. This fact is known as RISE (Shimakawa et al., 2018). The authors did not test the possibility of RISE to regulate the activity of Mehler reaction. That is, the 16O_2_ evolution rate regulates 18O_2_-uptake rate in PSI. Therefore, the authors should show 16O_2_ evolution rate in the mutants used in the present manuscript.

From these points, the reviewer does not recommend the present manuscript for the publication in *eLife*.

References

1) Reduction-Induced Suppression of Electron Flow (RISE) in the Photosynthetic Electron Transport System of Synechococcus elongatus PCC 7942.

Shaku K, Shimakawa G, Hashiguchi M, Miyake C. Plant Cell Physiol. 2016 Jul;57(7):1443-1453. Epub 2015 Dec 26.PMID: 26707729

2) Reduction-Induced Suppression of Electron Flow (RISE) Is Relieved by Non-ATP-Consuming Electron Flow in Synechococcus elongatus PCC 7942.

Shimakawa G, Shaku K, Miyake C. Front Microbiol. 2018 May 7;9:886. doi:0.3389/fmicb.2018.00886. eCollection 2018. PMID: 29867800

3) Comparative analysis of strategies to prepare electron sinks in aquatic photoautotrophs.

Shimakawa G, Murakami A, Niwa K, Matsuda Y, Wada A, Miyake C. Photosynth Res. 2019 Mar;139(1-3):401-411. doi: 10.1007/s11120-018-0522-z. Epub 2018 May 29. PMID: 29845382

4)Oxidation of P700 Ensures Robust Photosynthesis.

Shimakawa G, Miyake C.Front Plant Sci. 2018 Nov 6;9:1617. doi: 10.3389/fpls.2018.01617. Collection 2018. Review. PMID: 30459798

5) Diversity of strategies for escaping reactive oxygen species production within photosystem I among land plants: P700 oxidation system is prerequisite for alleviating photoinhibition in photosystem I.

Takagi D, Ishizaki K, Hanawa H, Mabuchi T, Shimakawa G, Yamamoto H, Miyake C. Physiol Plant. 2017 Sep;161(1):56-74. doi: 10.1111/ppl.12562. Epub 2017 May 24. PMID: 28295410

*Reviewer #2:*

This is an important, comprehensive and interesting paper worth considering for publication in this journal. Using stable O_2_ isotopes and a MIMS the authors examined the routes of electrons in various mutants impaired in the 4 FLV proteins in a model cyanobacteria. These FLVs were shown to be involved in photoreduction of O_2_.

I do have several problems with the data presented in Figure 1. The cells hardly respired (higher under HC) during the first 4 minutes in the dark compared with that observed after darkening at the end of phase III. Is the rate of dark respiration (O_2_ uptake during the second dark phase in Figure 1) faster in flv4-2/OE mutant? And if so, why? This appear to be the case under both LC and HC.

What do you mean by 50% WT and why are the bands showing lower MW compared with the rest?

Why is phase II missing in mutant M55? Does it mean that electrons are re-routed in the NdhB mutant?. Can you check it in mutants ndhD1/3 vs ndhD2/4? I am not requesting this experiment but it may help understand the transient shape in the WT!!

In subsection “Extent and kinetics of the Mehler-like reaction in cells acclimated to low (LC) and high Ci (HC) conditions” we read "…confirms that the Flv1 and Flv3 proteins are responsible for the Mehler-like reaction under HC condition…" I don't see these data here, is it correct the Flv1 and Flv3 contributes under HC only?

Subsection “Extent and kinetics of the Mehler-like reaction in cells acclimated to low (LC) and high Ci (HC) conditions”: you are most probably correct but unlike mutant M55 the HC grown WT cells possess some NDH1, isn't it? It is able to utilize glucose, unlike M55.

Subsection “The effect of sodium carbonate”: How much sodium carbonate was used under the various experiments including the pH conditions? Altogether, I am confused regarding the concentration of sodium under the various growth conditions. Note that sodium is essential for two of the main bicarbonate transport systems. In fact, if sufficient bicarbonate is used the cells behave like HC-grown. Thus your finding that transcript is unaffected is quit surprising unless the concentration of carbonate used was negligible. Please make sure to provide the amounts of carbonate added in each of the experiments! throughout!!

Any reason the growth rates (as depicted in Figure 4) is that slow

Subsection “The FDP induced O_2_ photoreduction does not occur at PSII or the PQ-pool level”: With the current technologies you did the best one can but we must remember that when you block one electron route they are channeled to the others. In other words, you can't really assess the rate of electron transport in a specific tube of this multichannel system by blocking anyone of them either by a mutation or by most inhibitors.

Figure 5: Do you possess a similar data set for cells grown and maintained under HC?

To the best of my knowledge, this is the first report where the double flv1/3 mutant shows light-dependent O_2_ uptake.

Subsection “Growth history of cells has long-term consequences on the Mehler-like reaction”: It is not surprising that the growth conditions strongly affected the data but what you really show is that the cells you used were not fully acclimated to LC, with serious implications on the interpretations of the data!! in other words the experiments were conducted under wrong growth conditions? How much bicarbonate was present in the experiment presented in Figure 6?

I am a bit lost regarding the discrepancy between the in vivo and in vitro data. When expressed in *E. coli* FLV3 shows NADPH-dependent O_2_ uptake. Since this is a 4 electron business it has got to be a homodimer. But in vivo, a flv1 mutant is essentially unable to show light-dependent O_2_ uptake. Why can't it form the homodimer in vivo? Can FLV2 or 4 substitute for Flv1 in flv1 or flv3 mutants? Apparently not. Do you see a conservative NADPH binding motif in FLV2 or 4 or does it rely on NADH under LC?

What sets the kinetic limitation on the FLVs to avoid NADPH dissipation and hence inhibition of CO_2_ fixation? Any idea?

How about the possibility that the cyanobacterial PGR5 function differently that in green algae? And that it indeed plays a role here? Do you possess the needed mutant to be placed on the MIMS?

Is it possible that under the redox pressure in cells experiencing CO_2_ limitation a cyclic ET is activated within PSII?

*Reviewer #3:*

The present manuscript investigated the function of flavodiiron proteins (FDPs) in *Synechocystis* sp. PCC 6803. This organism possesses four FDPs (Flv1-4) essential for photoprotection of photosynthesis. The authors conclude that Flv2/Flv4 contribute to the Mehler-like reaction under defined conditions. Moreover, it is suggested that O_2_ photoreduction via Flv2/Flv4 occurs down-stream of PSI in a coordinated manner with Flv1/Flv3. The data as provided by the authors appear to support their conclusions.

However, a fundamental problem with this manuscript is that only O_2_ uptake rates are provided. As rate and extent of O_2_ photoreduction/uptake is also dependent on the capacity of light-driven photosynthetic electron transfer leading to O_2_ evolution, also O_2_ evolution rates need to be specified for the different conditions and mutants by using MIMS. For the moment it is not possible to exclude that differences observed in O_2_ photoreduction rates are simply due to difference in capacity of photosynthetic electron transfer, which in turn affects O_2_ uptake.

It would be important to determine the ratio of O_2_ evolution versus O_2_ uptake for the different conditions and mutants.

[Editors’ note: what now follows is the decision letter after the authors submitted for further consideration.]

Thank you for submitting your article "Flavodiiron proteins 1-to-4 function in versatile combinations in O_2_ photoreduction in cyanobacteria" for consideration by *eLife*. Your manuscript has been reviewed by two peer reviewers, and the evaluation has been overseen by a Reviewing Editor and Detlef Weigel as the Senior Editor. The following individuals involved in review of your submission have agreed to reveal their identity: Michael Hippler (Reviewer #3).

The reviewers have discussed the reviews with one another and the Reviewing Editor has drafted this decision to help you prepare a revised submission.

The reviewers are still very reserved and asked for further clarifications. They are particularly worried about the low O_2_ evolution rate of M55 and propose O_2_ evolution measurement in phase1 to substantiate the drawn conclusions. They are also generally concerned about the M55 mutant background, because the original mutant cannot grow under low C O_2_ levels, but does so under your condition.

Please, have also a look at the specific reviewer comments below, which may guide you to revise your manuscript.

*Reviewer #2:*

You may remember/note that my criticism of the earlier version was the modest among the three reviewers and that I recommended to accept their appeal. But when I started reading the new version I came across, in the fifth paragraph of the Results, that mutant M55 was grown under low level of CO_2_. This isn't a typo error – in their reply to the reviewers we read:

Q: You are most probably correct but unlike mutant M55 the HC grown WT cells possess some NDH1, isn't it? It is able to utilize glucose, unlike M55.

A: Yes, that is right, we are sorry for non-clarity. We have modified the sentence.

This mutant, originally raised by Teruo Ogawa many years ago, was characterized by its high CO_2_ requiring phenotype. It can only grow under very high CO_2_ level. One may grow it under high CO_2_ and then exposed to a lower one, this is fine. This could be the case in the earlier version though not define, but this is clearly not the case here. Possibly, under the stressing conditions it went through suppression mutations and is now able to grow under low CO_2_ level. You may ask the authors whether it is also able to grow on glucose (the original M55 mutant can't). The authors must re-sequence the mutant and find the suppression mutation. We can't proceed evaluating the manuscript unless we know what the nature of the mutation was and unless they compare the data shown here with those obtained in a high-CO_2_ requiring M55 mutant, they may have to construct it again.

Further, unlike earlier studies it is proposed that FLV2 and 4 take part in light-dependent O_2_ uptake. In view of the possibility that their growth conditions promoted accumulation of secondary mutations I examined the sequences of flv2 and 4 in cyanobase to find that it is unlikely that NADPH could bind there, questioning their conclusion. I am not clear whether they actually tested NADPH-dependent O_2_ uptake in vitro, shown to be very low in a study from another group. Given the in vitro results in the literature it is unlikely that FLV2 and 4 really take part. If they do I would test the sequence here too.

Altogether, while I can't recommend further assessment of the present version I strongly recommend resequencing of M55 mutant and their WT, it may end up with exciting results.

*Reviewer #3:*

Some aspects regarding the revised manuscript need to be addressed.

In Figure 1—source data 2, column G, the gross O_2_ uptake rate (µmol O2.mg Chla -1.hr-1) is shown. In the manuscript the authors refer to the gross O_2_ evolution rate, although this is not shown in this figure. Is it possible that instead of gross O_2_ uptake uptake, gross O_2_ evolution rates are shown in column G? This needs to be clarified.

Notable, the M55 mutant has a significantly reduced gross O_2_ evolution rate (considering, column G as gross O_2_ evolution rate) compared to WT, thus changes in O_2_ uptake rate might be linked to electron transfer capacity. Here it would be important to show O_2_ evolution in phase1.

HQNO also blocks b6f complexes. This should be considered in the interpretation of the results.

Looking at gross O_2_ evolution rates (considering, column G as gross O_2_ evolution rate) for Figure 5, ∆flv1/∆flv3 as well as ∆flv4 have a significant lower O_2_ evolution rates under 1000 and 1500 uE despite the fact that O_2_ uptake is compromised. Thus indicating that the absence of ∆flv1/∆flv3 as well as of ∆flv4 impacts O_2_ evolution. This should be considered in the respect of differences in O_2_ uptake and in regard to protection of the photosynthetic machinery. In this line, also data for PSII and PSI functionality at the different light intensities should be presented.

---

## [Author Response]

The editorial decision letter and the comments of reviewer #1 and reviewer #3 clearly show that the main reason for a rejection of the manuscript is the absence of gross O_2_ production data. In fact, we have simultaneously monitored the O_2_ uptake and O_2_ production during all the MIMS experiments. We do have figures done depicting the O_2_ production data, which can be immediately incorporated into the manuscript. I would like to mention that, contrary to the assumptions of the reviewer #1, the O_2_ evolution data will not change the main conclusions and interpretation of the presented results, since no dramatic differences were observed between the wild-type and the FDP mutants. This was the main reason why we didn’t present the O_2_ evolution data, and instead, tried to focus primarily on the Mehler-like reaction. However, now we agree that presenting the gross O_2_ production would be necessary for a validation of the presented O_2_ uptake results.

[Editors’ note: the author responses to the first round of peer review follow.]

Reviewer #1:

The authors analyze the functions of FLVs to catalyze Mehler reaction in S.6803 using MIMS. The reviewer cannot find any novel thing of FLVs in their physiological functions. Helman et al. (2003) already reported that FLVs catalyze O2-dependent electron flow in vivo using MIMS. Furthermore, the authors in the present manuscript already reported the functions of FLV2/4 to catalyze Mehler reaction.

This is a wrong statement. There is no in vivo evidence on Flv2/Flv4 contribution to O_2_ photoreduction (Mehler-like reaction) in Helman et al., 2003. Moreover, we have not published nor hypothesized that Flv2/4 could be involved in the Mehler-like reaction (if this is what the reviewer criticizes with his confusing comment above “…the authors in the present manuscript already reported the functions of FLV2/4 to catalyze Mehler reaction”, and what we indeed are reporting now in the present manuscript). As to the Helman et al. paper, it describes O_2_ photoreduction in experimental cultures grown under high CO_2_ level (HC, 5% CO_2_ bubbled in air), where the expression of the *flv4-flv2* operon is strongly downregulated (Zhang et al., 2009; Hackenberg et al., 2012, and Figure 1C in the present paper). Applying proper growth conditions (cultures grown for 4 days under air levels of CO_2_) and using different Flv mutants in this work we provide for the first-time evidence indicating the in vivo contribution of Flv2/Flv4 heterodimer to the Mehler-like reaction in *Synechocystis* sp. PCC 6308.

As mentioned in our manuscript, the group of Miyake has published a paper suggesting that Flv4 catalyzes O_2_ uptake but provided no direct evidence (Shimakawa et al., 2015). The suggestion was based on in vitro assay which demonstrated only residual O_2_ uptake activity of the recombinant Flv4 (20 min^-1^*vs* 40 s-1 from *Giardia*) and showed no affinity to NADPH but only to NADH.

As an important, serious issue, the authors did not show any data for 16O_2_ evolution in PSII, which reflects the electron flux in photosynthetic linear electron flow in vivo. The reviewer cannot understand why the authors did not show these data. The 162-evolution rate greatly affects photosynthetic electron transport rate. Miyake'group already showed the deficiency of FLV suppresses photosynthetic linear electron transport rate, and also suppresses CO_2_-dependent photosynthesis rate (Shaku et al., 2016). The deficiency of FLV induces the reduction of plastoquinone pool, which suppresses Q-cycle. This fact is known as RISE (Shimakawa et al., 2018). The authors did not test the possibility of RISE to regulate the activity of Mehler reaction. That is, the 16O_2_ evolution rate regulates 18O_2_-uptake rate in PSI. Therefore, the authors should show 16O_2_ evolution rate in the mutants used in the present manuscript.

We agree with both reviewers (reviewer #1 and #3) that the presentation of the gross O_2_ production would be necessary to show for validation of the O_2_ reduction results. In fact, we always monitor simultaneously the O_2_ production and O_2_ consumption rates during all the MIMS experiments. Since no drastic differences in the O_2_ evolution rates were observed between the wild-type and the Flv mutants, we did not present the O_2_ production data, and instead, tried to focus primarily on the Mehler-like reaction. However, we completely agree with the reviewers that, even though the O_2_ production rates were found not to change, this data should be added to the manuscript in order to facilitate the interpretation of obtained O_2_ photoreduction data and the conclusions made in the manuscript. We have now added this data and discuss the gross O_2_ values in the WT and the Flv mutant strains (Results section and the Figure 1—source data 2).

*About RISE*. We have used, as reviewer # 2 stated, “the best available technique” to study the Mehler-like reaction. The in-direct methods (*e.g*. RISE) used by Miyake team, cannot precisely address the scientific question related to the Mehler-reaction and, thus, are not commonly accepted by the scientific community. As an example “the RISE method” suggested by the Miyake team (Shaku et al., 2016)has so far been cited in 8 PubMed articles, and 7 of them are self-citations.

Reviewer #2:

This is an important, comprehensive and interesting paper worth considering for publication in this journal. Using stable O_2_ isotopes and a MIMS the authors examined the routes of electrons in various mutants impaired in the 4 FLV proteins in a model cyanobacteria. These FLVs were shown to be involved in photoreduction of O_2_.I do have several problems with the data presented in Figure 1. The cells hardly respired (higher under HC) during the first 4 minutes in the dark compared with that observed after darkening at the end of phase III. Is the rate of dark respiration (O_2_ uptake during the second dark phase in Figure 1) faster in flv4-2/OE mutant? And if so, why? This appear to be the case under both LC and HC.

That’s right, higher post-illumination O_2_-uptake in the dark is a well-known phenomenon and can be observed also with Clark-type electrode during the measurements of O_2_ exchange in WT. Similar phenomenon occurs in all the studied Flv mutants, including *flv4-2*OE. This might be related to a light-induced activation of the respiratory pathway. However, the current work focuses on the activity of Flvs in O_2_ reduction occurring during illumination thus, the increase of post-illumination O_2_ uptake rate was not discussed in the text.

What do you mean by 50% WT and why are the bands showing lower MW compared with the rest?

In Figure 1C, “50% WT” corresponds to 1:2 diluted WT total protein sample (20 μg of total protein is normally loaded (100%) and 50% correspond to 10 μg total protein loaded). The dilution of protein samples is included in all the immunoblot figures (Figure 1, Figure 3 and 6B) to show the linear range of the detection for the specific antibodies used. Slightly lower apparent MW is related to less protein loaded. Now we have clarified this point in the corresponding Figure legends.

Why is phase II missing in mutant M55? Does it mean that electrons are re-routed in the NdhB mutant?. Can you check it in mutants ndhD1/3 vs ndhD2/4? I am not requesting this experiment but it may help understand the transient shape in the WT!!

In the current work we hypothesize that during illumination, there is a competition between Flvs and the NDH-1 complex for the photosynthetic electrons. We speculate that the decay of O_2_ photoreduction rates in phase II corresponds to the redirection of electrons towards NDH-1, which consequently, results in a decrease of the Flv-driven O_2_ photoreduction. This hypothesis is supported by the constant O_2_ photoreduction in the M55 mutant, where no active NDH-1 complex competes for electrons with the Flvs, thus the Flv-driven O_2_ photoreduction does not decay during illumination and no phase II is observed. We have rephrased the text in Results section paragraph four to facilitate the reading. Please also see subsection “Under LC, the Flv1/Flv3 heterodimer is a rapid, strong and transient electron sink whereas Flv2/Flv4 supports steady-state O2 photoreduction”).

The ∆*ndhD1/D2* mutant, lacking the NDH-1L complex involved in CET, shows only a small decay (phase II) followed by constant O_2_ photoreduction (phase III) during illumination of 5 min or 10 min. The data is not included in the paper and is a subject of a new manuscript.

In subsection “Extent and kinetics of the Mehler-like reaction in cells acclimated to low (LC) and high Ci (HC) conditions” we read "…confirms that the Flv1 and Flv3 proteins are responsible for the Mehler-like reaction under HC condition…" I don't see these data here, is it correct the Flv1 and Flv3 contributes under HC only?

Flv1/3 contributes to Mehler reaction also under low CO_2_, being responsible for the fast O_2_ photoreduction. *Synechocystis* cells grown under high CO_2_ levels (HC) have shown suppressed expression of the *flv4-flv2* operon and strong downregulation of *flv3* at the transcript and protein levels (Zhang et al., 2009; Hackenberg et al., 2012 and Figure 1C in this work). Therefore, in the absence of Flv2/Flv4 heterodimer, the constant O_2_ photoreduction rate observed in the WT cells grown under high CO_2_ will be mainly carried out by the small amount of Flv1/Flv3 heterodimers (decreased Flv3 accumulation in HC compared to LC conditions is shown in Figure 1C). This is supported by the drastic decrease of O_2_ photoreduction in the ∆*flv1/3* and ∆*flv3/4* mutants grown under HC conditions (see Figure 1—figure supplement 1). We have slightly reformulated the text to facilitate the reading (see subsection “Extent and kinetics of the Mehler-like reaction in cells acclimated to low (LC) and high Ci (HC) conditions”).

Subsection “Extent and kinetics of the Mehler-like reaction in cells acclimated to low (LC) and high Ci (HC) conditions”: you are most probably correct but unlike mutant M55 the HC grown WT cells possess some NDH1, isn't it? It is able to utilize glucose, unlike M55.

Yes, that is right, we are sorry for non-clarity. We have modified the sentence.

Subsection “The effect of sodium carbonate”: How much sodium carbonate was used under the various experiments including the pH conditions? Altogether, I am confused regarding the concentration of sodium under the various growth conditions. Note that sodium is essential for two of the main bicarbonate transport systems. In fact, if sufficient bicarbonate is used the cells behave like HC-grown. Thus your finding that transcript is unaffected is quit surprising unless the concentration of carbonate used was negligible. Please make sure to provide the amounts of carbonate added in each of the experiments! throughout!!

Thank you for the comment. In this work we used standard BG11 medium containing sodium carbonate (Na_2_CO_3_) at a final concentration of 0.189 mM for all experimental cultures. Only when indicated, strong limitation of inorganic carbon was achieved by omitting Na_2_CO_3_ from the BG11 media. This is now indicated in the Figure legends and in the main text more clearly. Sodium nitrate (NaNO_3_) macronutrient is the main source of sodium (Na^+^) in BG11 media and it is present at a final concentration of 17.6 mM in all experimental cultures in this work. Therefore, the cultures grown in BG11 media without sodium carbonate should not experience strong limitation of sodium.

In this work, we demonstrate that the concentration (at LC and HC conditions) and distribution of dissolved inorganic carbon species (DIC) in the medium (dependent on the pH) are essential factors determining the expression and/or function of Flv proteins in *Synechocystis* sp. PCC 6308. The stronger limitation of inorganic carbon by omitting 0.189 mM sodium carbonate from cultures grown in BG-11 at pH 7.5 under air levels of CO_2_ affected neither the expression nor accumulation of Flv proteins, however, this limitation was enough to enhance the O_2_ photoreduction activity by the Flv proteins, as shown in Figure 2.

We agree that the amounts of carbon must be repeated in each Figure legend to make it easy to read and follow the manuscript. As recommended, we have mentioned the concentration of sodium carbonate present in the BG11 medium used in this work throughout the text.

Any reason the growth rates (as depicted in Figure 4) is that slow

Figure 4A and 4B depict growth at low light background (20 μ µmol photons m^-2^ s^-1^ background light regularly punched with 30 sec 500 μ µmol photons m^-2^ s^-1^ pulse) therefore slow growth is not a surprise. At 50 μ µmol photons m^-2^ s^-1^ background the cells grow “normally” (at least in our lab conditions with our WT cells) reaching OD_750_ = 2 after 4 days at pH 9. The slow growth observed at pH 6 is due to the low solubility of CO_2_ at acidic pH.

Subsection “The FDP induced O_2_ photoreduction does not occur at PSII or the PQ-pool level”: With the current technologies you did the best one can but we must remember that when you block one electron route they are channeled to the others. In other words, you can't really assess the rate of electron transport in a specific tube of this multichannel system by blocking anyone of them either by a mutation or by most inhibitors.

We completely agree with the reviewer. The closest we can get to reality with deletion mutants is that we assess the replaceability of certain electron transport routes and redox components. The more profound the effect, the more important the component is. With the right setup, we can map which pathway substitutes for the deleted one and determine a condition-specific hierarchy.

Figure 5: Do you possess a similar data set for cells grown and maintained under HC?

According to the suggestion of the reviewer, we have performed a new experiment with HC grown cells. The data is added to Figure 5—figure supplement 1.

To the best of my knowledge, this is the first report where the double flv1/3 mutant shows light-dependent O_2_ uptake.Subsection “Growth history of cells has long-term consequences on the Mehler-like reaction”: It is not surprising that the growth conditions strongly affected the data but what you really show is that the cells you used were not fully acclimated to LC, with serious implications on the interpretations of the data!! in other words the experiments were conducted under wrong growth conditions?

Not only acclimation to low carbon but also light intensity plays a role in accumulation level of Flvs. Rather than “wrong growth conditions”, we consider that in our previous studies, the accumulation of Flv2 and Flv4 in the cells was not enough to detect their O_2_ photoreduction activity using our MIMS setup. In the current work, we were able to induce higher accumulation of Flvs by reducing the inoculum size of the experimental cultures (allowing more light penetration) and extending the LC acclimation to 4 days. The increased accumulation of Flvs in the cells allowed us to demonstrate, for the first time, that O_2_ photoreduction is carried out by Flv2 and Flv4 proteins in the ∆*flv1/3* mutant cells. We have reformulated the text to facilitate the reading.

How much bicarbonate was present in the experiment presented in Figure 6?

All the growth experiments were done in standard BG-11 medium containing sodium carbonate (Na_2_CO_3_) at a final concentration of 0.189 mM (including the cells studied in Figure 6). Prior to the MIMS measurements (including Figure 6), the cells were supplemented with 1.5 mM of sodium bicarbonate (NaHCO_3_). This was important to ensure comparable conditions between measurements with MIMS. Independent experiments performed on WT cells grown in the absence of Na_2_CO_3_, but supplied with 1.5 mM NaHCO_3_ prior to the MIMS measurement showed no significant difference in O_2_ photoreduction rates (see the Figure 1—figure supplement 2), and thus should not affect the interpretation of the results. As recommended, the concentration of carbonate/bicarbonate is mentioned throughout the text (see legend of all Figures (1-6).

I am a bit lost regarding the discrepancy between the in vivo and in vitro data. When expressed in E. coli FLV3 shows NADPH-dependent O_2_ uptake. Since this is a 4 electron business it has got to be a homodimer. But in vivo, a flv1 mutant is essentially unable to show light-dependent O_2_ uptake. Why can't it form the homodimer in vivo? Can FLV2 or 4 substitute for Flv1 in flv1 or flv3 mutants? Apparently not. Do you see a conservative NADPH binding motif in FLV2 or 4 or does it rely on NADH under LC?

Two in vitro studies have been performed on recombinant *Synechocystis* Flv3 or Flv4 proteins purified after expression in *E. coli* (Vicente et al., 2002, Shimakawa et al., 2015). Both studies claim that Flv3 (Vicente et al., 2002) or Flv4 (Shimakawa et al., 2015) function as a homo-dimer in O_2_ reduction. However, in both cases the enzyme activity is only residual (0.38 min^-1^ for Flv3, 20 min^-1^ for Flv4 compared to 40 s-1 of the Flv activity in *Giardia*). Moreover, studies have shown either little (Vicente et al., 2002) or no affinity (Shimakawa et al., 2015) to NADPH, a reducing equivalent produced by photosynthetic linear electron transport. Many years of work in our laboratory and in several other laboratories have shown that the isolation of functional Flvs from photosynthetic organisms is an extremely difficult task and so far none of the laboratories has managed to isolate highly active Flvs from photosynthetic organisms. Therefore, we believe that in vitro experiments were not reflecting a real situation. in vivo experiments clearly demonstrate that homooligomers cannot function in O_2_ uptake (Mustila et al., 2016).

Since, no crystal structures of the Flvs from oxygen evolving photosynthetic organisms are available (except recent structure of the truncated Flv1 lacking a flavodoxin-like domain, Borges et al., 2019), it is not possible to confidently conclude about the NADH or NAPDH preference just based on the amino acid sequence and homology searches.

What sets the kinetic limitation on the FLVs to avoid NADPH dissipation and hence inhibition of CO_2_ fixation? Any idea?

This is a key question and so far, an unanswered one. One possibility is post-transcriptional modification of the protein, such as phosphorylation (Angeleri et al., 2016) and/or *pmf* based regulation.

How about the possibility that the cyanobacterial PGR5 function differently that in green algae? And that it indeed plays a role here? Do you possess the needed mutant to be placed on the MIMS?

This is a logical question. In future we will probe also this mutant. However, in this work we concentrate on dissection of Flv1/3 and Flv2/4-originated O_2_ photoreduction and Ci-dependency of O_2_ photoreduction.

Moreover, *Synechocystis* Pgr5 deletion mutant, different from *Chlamydomonas* Pgr5 knockout mutant (Jokel et al., 2018), do not show any phenotype under fluctuating light intensities (Allahverdiyeva et al., 2013). Therefore we assume that in *Synechocystis* a strong crosstalk between these proteins is missing, at least under conditions studied so far.

Is it possible that under the redox pressure in cells experiencing CO_2_ limitation a cyclic ET is activated within PSII?

Unfortunately, we cannot answer this interesting question.

Reviewer #3:

The present manuscript investigated the function of flavodiiron proteins (FDPs) in Synechocystis sp. PCC 6803. This organism possesses four FDPs (Flv1-4) essential for photoprotection of photosynthesis. The authors conclude that Flv2/Flv4 contribute to the Mehler-like reaction under defined conditions. Moreover, it is suggested that O_2_ photoreduction via Flv2/Flv4 occurs down-stream of PSI in a coordinated manner with Flv1/Flv3. The data as provided by the authors appear to support their conclusions.However, a fundamental problem with this manuscript is that only O_2_ uptake rates are provided. As rate and extent of O_2_ photoreduction/uptake is also dependent on the capacity of light-driven photosynthetic electron transfer leading to O_2_ evolution, also O_2_ evolution rates need to be specified for the different conditions and mutants by using MIMS. For the moment it is not possible to exclude that differences observed in O_2_ photoreduction rates are simply due to difference in capacity of photosynthetic electron transfer, which in turn affects O_2_ uptake.It would be important to determine the ratio of O_2_ evolution versus O_2_ uptake for the different conditions and mutants.

We agree with the reviewer that the presentation of the gross O_2_ production would be necessary for validation of the O_2_ photoreduction results. We have simultaneously monitored both the O_2_ photoreduction and O_2_ production during all the MIMS experiments and the data have been added to the revised manuscript. Importantly, no significant difference in the gross O_2_ evolution rates was observed between the wild-type and the FDP mutants.

[Editors’ note: the author responses to the re-review follow.]

The reviewers are still very reserved and asked for further clarifications. They are particularly worried about the low O_2_ evolution rate of M55 and propose O_2_ evolution measurement in phase1 to substantiate the drawn conclusions. They are also generally concerned about the M55 mutant background, because the original mutant cannot grow under low CO_2_ levels, but does so under your condition.Please, have also a look at the specific reviewer comments below, which may guide you to revise your manuscript.

Reviewer #2:

You may remember/note that my criticism of the earlier version was the modest among the three reviewers and that I recommended to accept their appeal. But when I started reading the new version I came across, in the fifth paragraph of the Results, that mutant M55 was grown under low level of CO_2_. This isn't a typo error – in their reply to the reviewers we read:Q: You are most probably correct but unlike mutant M55 the HC grown WT cells possess some NDH1, isn't it? It is able to utilize glucose, unlike M55.A: Yes, that is right, we are sorry for non-clarity. We have modified the sentence.This mutant, originally raised by Teruo Ogawa many years ago, was characterized by its high CO_2_ requiring phenotype. It can only grow under very high CO_2_ level. One may grow it under high CO_2_ and then exposed to a lower one, this is fine. This could be the case in the earlier version though not define, but this is clearly not the case here. Possibly, under the stressing conditions it went through suppression mutations and is now able to grow under low CO_2_ level. You may ask the authors whether it is also able to grow on glucose (the original M55 mutant can't). The authors must re-sequence the mutant and find the suppression mutation. We can't proceed evaluating the manuscript unless we know what the nature of the mutation was and unless they compare the data shown here with those obtained in a high-CO_2_ requiring M55 mutant, they may have to construct it again.

We understand the concern of the reviewer 2 related to the background of the strains. Indeed, in the original paper it was shown that M55 cannot grow under air level CO_2_ (Ogawa T. 1991 “A gene homologous to the subunit-2 gene of NADH dehydrogenase is essential to inorganic carbon of *Synechocystis* PCC 6803”, PNAS 88:4275–427). However, in this paper the growth experiment was performed with very diluted cell culture (OD_730_=0.03) and at relatively high light intensity (120 umol m^-2^ s^-1^). It is not surprising that M55, the mutant strain deficient in important bioenergetic processes such as C_i_ assimilation, respiration and cyclic electron transfer, shows strong growth retardation under such stressful condition.

Most importantly, the same author later published 2 papers where he precisely described the growth characteristics of the M55mutant. In both papers it is clearly shown that the M55 mutant is able to grow (at a slower rate compared to WT) under air level CO_2_ conditions (pH 8.0) in liquid medium and on BG-11 agar plates under moderate light intensity of 60 umol m^-2^ s^-1^ (see Ohkawa, Price, Badger, Ogawa “Mutation of *ndh* genes leads to inhibition of CO_2_ uptake rather than HCO3−uptake in *Synechocystis* sp. strain PCC6803”, 2000, Journal of Bacteriology p. 2591–2596, in Figure 3 and Figure 4; Ohkawa, Pakrasi, Ogawa ‘Two types of functionally distinct NAD(P)H Dehydrogenases in *Synechocystis* sp. strain PCC6803’ 2000 JBC 275, p. 31630-4; in Figure 1).

There are a bunch of papers where M55 was grown under air level CO_2_, high pH conditions. In our lab, we routinely cultivate M55 at pH 8.2 at air level CO_2_ (e.g. Zhang et al., 2004; Zhang et al., 2012).

Following the suggestion of the reviewer 2, we performed a growth trial with M55 in the presence of 5mM glucose + 10uM DCMU. As expected, *M55 was unable to grow under photoheterotrophic conditions* (Author response image 1), which is in line with the previous observations of photoheterotrophic growth deficiency of M55 (Ohkawa et al., 2000b in Figure 1, Zhao et al., 2015 in Figure 5). Moreover, we have cultivated the M55 strain at air-level CO_2_ at pH 8.2 and we were able to reproduce the results published in Ohkawa et al., 2000*a*, where M55 demonstrates slower growth compared to wildtype (Author response image 1).

**Author response image 1. respfig1:** Photoheterotrophic growth of wild-type (WT) and M55 cells on BG-11 agar plates. The WT and M55 mutant cells of *Synechocystis* were resuspended in BG-11 median at pH8.2. Three microliters of cell suspensions at OD_750nm_ of 0.1 (top row), 0.01 (middle row), and 0.001 (bottom row) was spotted on agar plates. Five millimolars of Glucose and 10 μm DCMU were added to the plates for photoheterotrophic growth (right side) or were not added for photoautotrophic growth (left side). The plates were grown under ambient air for 8d at 50 μmol photons m^-2^ S^-1^.

In light of this, we respectfully disagree that resequencing of M55 mutant is necessary for this manuscript. Considering the fact that M55 was generated decades ago, there is no doubt that many other mutations have likely occurred during the cultivation but in respect of Ci assimilation and respiration, which is the focus of this manuscript, we still observe the original phenotype.

Further, unlike earlier studies it is proposed that FLV2 and 4 take part in light-dependent O_2_ uptake. In view of the possibility that their growth conditions promoted accumulation of secondary mutations I examined the sequences of flv2 and 4 in cyanobase to find that it is unlikely that NADPH could bind there, questioning their conclusion. I am not clear whether they actually tested NADPH-dependent O_2_ uptake in vitro, shown to be very low in a study from another group. Given the in vitro results in the literature it is unlikely that FLV2 and 4 really take part. If they do I would test the sequence here too.

The reason why earlier studies did not link the Flv2 and Flv4 proteins to the light-induced O_2_ uptake is simple: the Flv2 and Flv4 deletion mutants were not probed by MIMS (previous works from our lab) or high CO_2_ grown mutant cells were used in the experiments (Helman et al., 2003). Since, under high CO_2_ conditions the *flv4-sll0218-flv2* operon is strongly downregulated, the similar phenotype between WT and the Flv2 and Flv4 deletion mutants is logical (Helman et al., 2003 and in this manuscript in Figure 1—figure supplement 1). We believe that re-sequencing here is unnecessary. Involvement of Flv2 and Flv4 in the light induced O_2_ uptake is solid: the ΔFlv2/4 lacks light-induced O_2_ uptake, whereas the Flv2/4 overexpression strain demonstrates significantly increased O_2_ uptake (see Figure 1 in this manuscript). In our lab we are well aware that secondary mutations frequently occur in cyanobacteria. In order to keep the original genome pool, we maintain our strains in cryogenic freezers, carefully avoid long-term handling and frequently start new cultures from the original cryo-stored stocks.

As reviewer pointed out “…it is proposed that FLV2 and 4 take part in light-dependent O_2_ uptake”. Indeed, this is one of the main messages of this paper and how we revealed the involvement of Flv2 and Flv4 in light-induced O_2_ uptake in vivo, is thoroughly discussed and explained. Therefore, we firmly believe that our observations do not require further supporting experiments.

Confidentconclusions about the absence or presence of NAD(P)H-binding sites, based solely on the amino acid sequence, are dubious. Well defined tertiary structural elements (see https://www.ebi.ac.uk/interpro/entry/IPR016040) form the NAD(P)H-binding motif and in the coordinative binding of NADPH, several amino acid residues take part all along the polypeptide. However, based on secondary structural element prediction and sequence conservation of NAD(P)H-binding proteins, attempts were made to predict NAD(P)H-binding (http://crdd.osdd.net/raghava/nadbinder/)and based on that, Flv4 might be able to bind NAD.

Importantly, both Flv4 and Flv2 conserve a Flavin reductase domain (IPR002563) at the C-termini, a domain found in NAD(P)H-flavin oxidoreductases. Moreover, the sequence similarity of Flv4 to Flv3 and of Flv2 to Flv1 (62% and 71% similarity, respectively) implies that Flv2/4 carries out similar reactions to Flv1/3. Besides, Flv3 and Flv1 conserve the same Flavin reductase domain as Flv2, and Flv4 (IPR002563) at the C-termini.

However, in the manuscript we do not make conclusion that Flv2/4 accepts electrons from NADPH. We hypothesize that Flv2/4 takes electrons from Fed or FNR (see Figure 7 in this manuscript) but for making such a strong conclusion, the in vitro assay with the recombinant proteins would be necessary. Unfortunately, we do not have functional recombinant protein (see our previous answer to the reviewers and the manuscript) therefore we cannot perform in vitro assays. Anyhow, the exact electron donor of Flv2/4 would only refine and not contradict our conclusions.

Reviewer #3:

Some aspects regarding the revised manuscript need to be addressed.In Figure 1—source data 2, column G, the gross O_2_ uptake rate (µmol O2.mg Chla -1.hr-1) is shown. In the manuscript the authors refer to the gross O_2_ evolution rate, although this is not shown in this figure. Is it possible that instead of gross O_2_ uptake uptake, gross O2 evolution rates are shown in column G? This needs to be clarified.

We thank the reviewer for pointing out this mistake. Indeed, there was a typo in the title of column G in Figure 1—source data 2. We have now corrected gross O_2_ uptake to gross O_2_ evolution.

Notable, the M55 mutant has a significantly reduced gross O_2_ evolution rate (considering, column G as gross O_2_ evolution rate) compared to WT, thus changes in O_2_ uptake rate might be linked to electron transfer capacity. Here it would be important to show O_2_ evolution in phase1.

This is a correct observation. Following the reviewer's suggestion, we have added gross O_2_ evolution data to Figure 1—source data 2. During the dark-to-light transition (Phase I), M55 demonstrates a slow induction of O_2_ evolution. Indeed, in M55, the light-induced O_2_ uptake is higher, whereas gross O_2_ evolution is notably lower, compared to WT grown under the same condition. This is mentioned in a revised manuscript.

HQNO also blocks b6f complexes. This should be considered in the interpretation of the results.

We thank the reviewer for this information. We have mentioned the inhibitory effect of HQNO to Cytb6f in the revised manuscript. However, the inhibitory effect of HQNO does not affect our conclusions, since it is used in combination with DBMIB, which is also a known inhibitor of Cyt*b*6*f*.

Looking at gross O_2_ evolution rates (considering, column G as gross O_2_ evolution rate) for Figure 5, ∆flv1/∆flv3 as well as ∆flv4 have a significant lower O_2_ evolution rates under 1000 and 1500 uE despite the fact that O_2_ uptake is compromised. Thus indicating that the absence of ∆flv1/∆flv3 as well as of ∆flv4 impacts O_2_ evolution. This should be considered in the respect of differences in O_2_ uptake and in regard to protection of the photosynthetic machinery. In this line, also data for PSII and PSI functionality at the different light intensities should be presented.

Involvement of Flv2 and Flv4 in photoprotection of PSII complex has been previously reported (Zhang et al., 2009, Hakkila et al., 2013, Bersanini et al., 2014, Chukhutsina et al., 2015, Bersanini et al., 2017), whereas the Flv1 and Flv3 proteins are linked mainly to photoprotection of PSI under fluctuating-light intensities (Allahverdiyeva et al., 2013). Following the reviewer's suggestion, we have performed new experiment, where the mutant strains grown at moderate light (50 µmol photons m^-2^s^-1^, same condition used in this work) were exposed to high-light (1500 µmol photons m^-2^ s^-1^) for 2 hours. PSII activity was probed as O_2_ evolution in the presence of artificial electron acceptor, DMBQ and PSI was evaluated as maximum oxidazable P700, Pm (Figure 5—figure supplement 2). In the revised manuscript we shortly mention about the photoprotective role of FLVs under high light intensities.